# Starch-mediated colloidal chemistry for highly reversible zinc-based polyiodide redox flow batteries

Zhiquan Wei[1], Zhaodong Huang[1,2], Guojin Liang[3] ✉, Yiqiao Wang[1], Shixun Wang [1], Yihan Yang[4], Tao Hu[5] & Chunyi Zhi [1,2,4] ✉

Aqueous Zn-I flow batteries utilizing low-cost porous membranes are promising candidates for high-power-density large-scale energy storage. However, capacity loss and low Coulombic efficiency resulting from polyiodide cross-over hinder the grid-level battery performance. Here, we develop colloidal chemistry for iodine-starch catholytes, endowing enlarged-sized active materials by strong chemisorption-induced colloidal aggregation. The size-sieving effect effectively suppresses polyiodide cross-over, enabling the utilization of porous membranes with high ionic conductivity. The developed flow battery achieves a high-power density of 42 mW cm$^{-2}$ at 37.5 mA cm$^{-2}$ with a Coulombic efficiency of over 98% and prolonged cycling for 200 cycles at 32.4 Ah L$^{-1}_{posolyte}$ (50% state of charge), even at 50 °C. Furthermore, the scaled-up flow battery module integrating with photovoltaic packs demonstrates practical renewable energy storage capabilities. Cost analysis reveals a 14.3 times reduction in the installed cost due to the applicability of cheap porous membranes, indicating its potential competitiveness for grid energy storage.

Energy storage is a vital technology to improve the utilization efficiency of clean and renewable energies, e.g., wind and solar energy, where the flow batteries with low-cost and high power are one of the most promising candidates for large-scale energy storage[1–5]. Aqueous zinc-iodine flow batteries (Zn-I FBs) hold great potential due to their intrinsic safety, high theoretical specific capacity (268 Ah L$^{-1}$), and high energy density[6–12]. However, the widely applied membrane is the expensive fluorinated Nafion-based membrane, leading to the relatively high overall cost of the battery[13,14]. Even though the dense Nafion-based membrane could endow high selectivity for improved coulombic efficiency (CE), it could be a trade-off in ion conductivity with increased membrane resistance to impair the voltage efficiency (VE) and energy efficiency (EE), especially at high working currents, thus the Zn-I FBs systems generally suffer from lower power density[15–19].

To solve these issues, modifications can be conducted on the membrane engineering and electrolyte regulations for Zn-I FBs, respectively[20,21]. Regarding the membrane, designing new polymeric membranes is retarded by the difficulties in endowing the membrane with precise pore-size modulation, controllable thickness and designated surface-charged states[14,20,22]. To mitigate it, low-cost polyolefin-based porous membranes (LPPM) have emerged as a promising separator to improve the working currents owing to the high ionic permeability with low ionic resistance. However, the pristine LPPM would inevitably accommodate low selectivity of the active materials, i.e., the charged iodine species, leading to the irreversibility at the cathode side with severe cross-over and capacity loss, as shown in Fig. 1a, b[23,24]. Meanwhile, the notorious cross-over issue of the catholyte would exacerbate, especially when Zn-I FBs are integrated with

[1]Department of Materials Science and Engineering, City University of Hong Kong, Hong Kong, China. [2]Hong Kong Center for Cerebro-Cardiovascular Health Engineering (COCHE), Hong Kong, China. [3]Faculty of Materials Science and Energy Engineering/Institute of Technology for Carbon Neutrality, Shenzhen Institute of Advanced Technology, Chinese Academy of Sciences (CAS) Shenzhen, Shenzhen, Guangdong, China. [4]Songshan Lake Materials Laboratory, Dongguan, Guangdong, China. [5]School of Materials Science and Engineering, Anhui University, Hefei, China. ✉e-mail: gj.liang@siat.ac.cn; cy.zhi@cityu.edu.hk

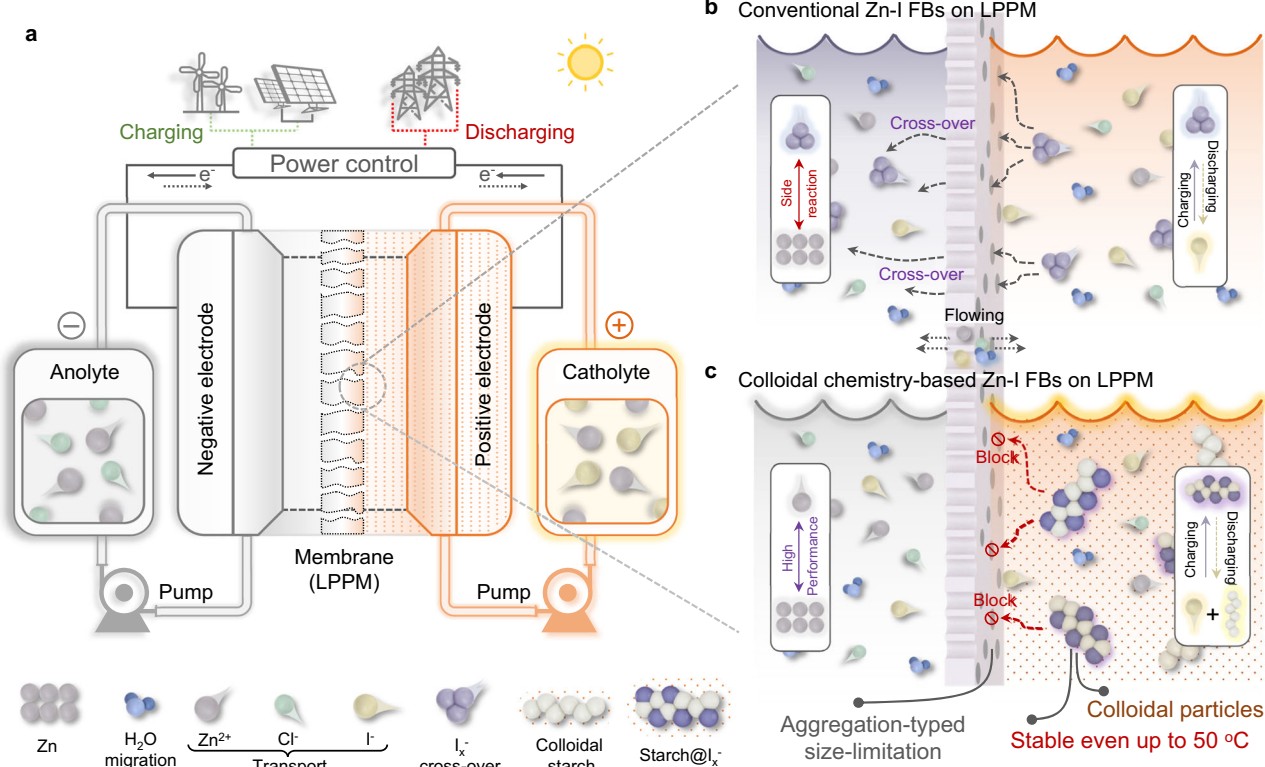

**Fig. 1 | Different processes in Zn-I FBs using a low-cost polyolefin-based porous membranes (LPPM) without/with colloidal starch. a** The schematic illustration of cross-over-free zinc-iodine flow batteries (Zn-I FBs) under room and high-temperature conditions. **b** Cross-over of polyiodide ($I_x^-$) through the pristine LPPM leads to a severe discharging cell of conventional Zn-I FBs. **c** Colloidal chemistry-based electrolytes restrict the cross-over of active materials ($I_x^-$) owing to the size limitation induced by the colloidal aggregation effect.

renewable power energy (i.e., solar and wind energy) and operating in outdoor environments with high temperatures[25–27]. Thereby, a functional coating layer can be generally introduced to the LPPM, featuring with ion-sieving limitation and/or charge repulsion to prevent the cross-over of the redox species[12,28]. However, such strategies would increase the ionic resistance and confront challenges such as the high cost and the precise synthesis of the nanostructured materials with the sieving sizes and the designated loaded charges.

Based on the principle of size-sieving effect in the LPPM, there is a significant potential to enhance the selectivity of those membranes by regulating the size of the active materials, i.e., the charged iodine species in the electrolyte. The target of the electrolyte regulation is to increase the size of the active iodine species, so that it can be blocked by the applied LPPM, eliminating the iodine species cross-over. Meanwhile, the size of the active iodine species should not be too large to induce low diffusion efficiency or even precipitation. Match between the size of active iodine species and pore of LPPM is a challenge and such regulation strategy has not been developed yet. Inspired by the strong interaction between low-cost starch and iodine, we reason that the tunable large-sized colloidal iodine-starch (IS) active species could possess great potential to avoid the loss of active redox caused by the LPPM, as schemed in Fig. 1c[29–33].

Herein, we developed the colloidal chemistry for iodine catholyte of Zn-I FBs via renewable and cost-effective starch. The IS-based active materials in the catholyte could be regulated to the aggregated colloidal nanoparticles, which are tuned to satisfy the size sieving rule to simultaneously achieve high ionic selectivity and ionic conductivity of the IS catholyte. The colloidal IS-based Zn-IS FBs with polypropylene (PP) membranes as LPPM could deliver superior performance of cycling stability for 350 cycles at high current density. In addition, due to the strong chemisorption between starch and iodine redox, the as-

developed colloidal IS systems remained stable. They could inhibit the cross-over issue even at high temperatures (50 °C) for stable cycling. Most significantly, benefiting from the utilization of LPPM and as-obtained high power, the installed cost of the 1-MW flow stack could dramatically decrease by 14.3 times compared to the installed cost for Nafion membranes.

## Results

### Characterizations of the colloidal electrolytes

Starch is a long-chain polymer of sugar molecules connected through glycosidic linkage, as shown in Supplementary Fig. 1[29]. The soluble amylose starch molecule is a linear polymer structure that can dissolve in water to form hydrogen bonds with water molecules and obtain a colloidal solution[30]. As displayed in Fig. 2a, the colloidal Tyndall effect can be observed in 1 M starch solution (M is molarity as mol L$^{-1}$), which indicates the nanosized colloidal starch ranging within 1–100 nm, benefiting from forming the colloidal aggregation. Compared to the bare $I_x^-$ solution without colloidal Tyndall effect, it could still be featured with the Tyndall effect as $I_x^-$ and starch-mixed solution in Fig. 2a. This result could indirectly demonstrate that the amylose-typed starch could capture the polyiodide anions $I_x^-$ by strong bonding interaction to form the starch/polyiodide colloidal complex[31].

To determine the suitable concentration of the starch-iodine electrolyte, the viscosity, ionic conductivity, and permeability of the starch-regulated electrolytes were investigated as the essential parameters to determine the overall performance of the full flow cell, such as VE and CE. As shown in Fig. 1b and Supplementary Fig. 2, the viscosity increased exponentially with increasing starch concentration, where the concentration of the ZnI$_2$ was fixed at 2 M. Specifically, the viscosity value of 1 M starch-containing solution is 13.5 kPa·s, which dramatically increased to 227 kPa·s of 2 M starch-containing

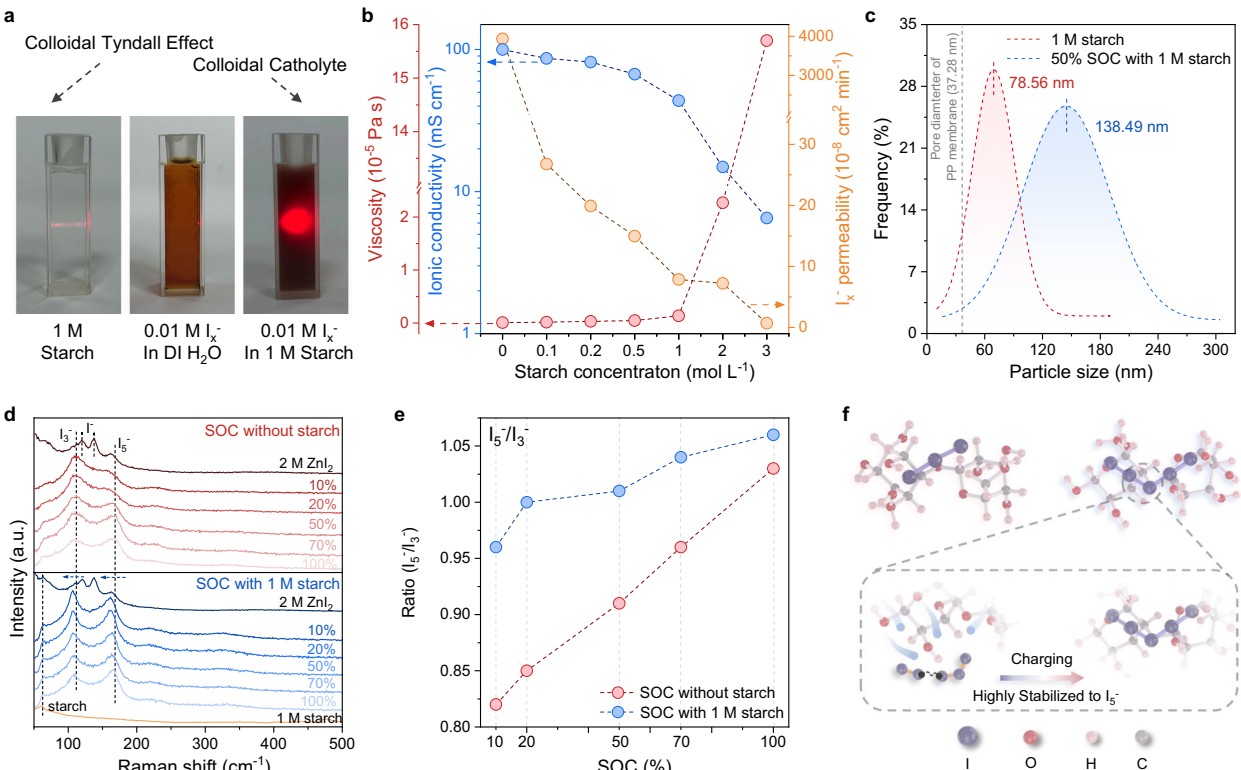

**Fig. 2 | Characterizations of starch-containing colloidal electrolytes. a** Digital images of blank starch solution (1 M), blank polyiodide ($I_x^-$) solution (0.01 M), and $I_x^-$ solution (0.01 M) with starch (0.1 M). **b** Viscosity, ionic conductivity, and $I_x^-$ permeability of 2 M $ZnI_2$ electrolytes with different starch concentrations (0, 0.1, 0.2, 0.5, 1, 2, and 3 M). **c** Size distribution of the colloidal particles in blank starch solution and starch at 50% SOC, respectively. **d** Raman spectra of starch/polyiodide complex at different SOCs and **e** corresponding calculated ratio between $I_5^-$ and $I_3^-$ of blank 2 M $ZnI_2$ and $ZnI_2$ with 1 M starch at different SOCs (10%, 20%, 50%, 70%, 100%). **f** The evolution of bonding energy of $I_3^-$ and $I_5^-$ interacting with the soluble starch.

electrolytes, i.e., 20 times larger than that of the 1 M starch. It was over 100 times higher than that of the 3 M starch electrolyte (1570 kPa·s). Along with the starch concentration increasing from 0 to 3 M, the ionic conductivity gradually decreases from 100.1 to 6.5 mS cm⁻¹. Generally, higher electrolyte viscosity would correspond to lower ionic conductivity, which indicates that the viscosity is the dominating factor in ionic conductivity, and 1 M starch in electrolytes can retain superior ionic conductivity. On the other side, two-compartment H-cells consisting of the IS catholyte at 50% state of charge in one cell and deionized water in another cell were used to evaluate the permeability of IS colloids across the PP membrane, as schemed in Supplementary Fig. 3. The UV-visible spectra and permeation rate of nominally prepared $I_x^-$ solutions under different starch concentrations (Supplementary Figs. 4–11), wherein the strong peaks with the absorption wavelength of 288 and 350 nm are attributed to the presence of $I_3^-$ [6,34]. In specific, the $I_x^-$ permeability largely decreased along with the increased starch concentration compared to the severe permeability in blank electrolytes, as shown in Fig. 2b. It indicated that the colloidal starch could strongly confine the iodine by forming a colloidal aggregation featuring low iodine permeability to impede the cross-over issue. Considering these observations, 1 M starch and 2 M $ZnI_2$-based IS electrolytes were selected as the prototypical electrolyte for the following investigations.

Next step, we have to ensure the sizes of nanoparticles in the colloidal aggregation match the pore size of polypropylene (PP) membranes as LPPM. The microscopic patterns and sizes were investigated by an atomic force microscope (AFM). Figure 2c and Supplementary Fig. 12 showed uniform nanoparticles of the bare starch colloid on the substrate with a distinguishable nano-size of ca. 78.56 nm (Supplementary Fig. 13). As displayed in Fig. 2c and Supplementary Figs. 14 and 15, it is seen that the size of the starch/polyiodide

colloids in 50% state of charge (ca. 138.49 nm) is larger than the pristine bare starch, which could be ascribed to the strong chemisorption to promote the aggregation between the colloidal starch molecules with abundant electron-rich hydroxyl groups and polyiodides $I_x^-$ [35–37]. Thus, following the size-sieving rule, it could effectively inhibit the permeability of IS colloids across PP membrane with nanosized pore diameters (Supplementary Fig. 16, ca. 37.28 nm), reasonably verifying that starch-containing colloidal electrolytes are beneficial for limiting the cross-over of active materials (polyiodides $I_x^-$) in the catholyte[38].

To understand the interaction behaviors between starch and iodine, the starch/polyiodide complex solutions in different state of charge (SOC) were further confirmed by Raman spectra (Fig. 2d, e). The Raman peaks at 120 and 137.16 cm⁻¹ can be ascribed to the skeletal vibrations of metal-iodine ions ($I^-$), while the two Raman peaks located at 110 and 160 cm⁻¹ can be attributed to the triiodide ion ($I_3^-$) and pentaiodide ion ($I_5^-$), respectively[31,39,40]. Regarding the electrolyte without starch, an intense $I_3^-$ signal along with a weak $I_5^-$ signal was shown in low 10% SOC and the $I_5^-$ species signal gradually increased as high content of active redox in high charging SOC. Based on the no-starch system in Fig. 2d, e, the $I_3^-$ peak exhibited barely shift but the $I_5^-$ peak became broaden with decreased intensity, which indicated the polyiodide complex was unstable especially at high concentrations. In contrast, both $I_3^-$ and $I_5^-$ species in starch-containing electrolytes presented stable peak signals and apparent blue shifts by increasing the SOC large ratio from 0% to 100% SOC. It indicated the colloidal starch could stabilize the polyiodide active species, i.e., $I_3^-$ and $I_5^-$, based on the strong chemical bonding for colloidal chemistry. Moreover, the stable existence of $I_5^-$ species on the colloidal starch can also be corroborated by *I 3d* XPS profiles as displayed in Supplementary Fig. 17, wherein the $I_5^-$ species presents a dominant role in starch-based electrolytes under 50% SOC compared with blank $ZnI_2$ electrolytes[40,41].

To understand in-depth the configurations of the IS species, density functional theory (DFT) calculations were conducted and ionic transference numbers were calculated to investigate the bonding energy of starch with iodine species ($I^-$, $I_3^-$ and $I_5^-$) and other charge carriers ($Zn_2^+$ and $Cl^-$). As displayed in Supplementary Fig. 18, the binding energies of different samples demonstrate the order of starch@$I^-$ ($E_b = -0.354$ eV) > starch@$Cl^-$ ($E_b = -0.119$) > starch@$Zn^{2+}$ ($E_b = -0.081$ eV). Furthermore, the $ZnI_2$ solution with colloidal starch exhibited a higher cation ($Zn^{2+}$) transference number ($t_+ = 0.651$) compared to the blank $ZnI_2$ solution ($t_+ = 0.484$), which can be attributed to the entrapment of anions ($I^-$) by starch (Supplementary Figs. 19 and 20). Conversely, the cation transference number of the $ZnCl_2$ solution with colloidal starch was only a marginal increase compared to that of the blank $ZnCl_2$ solution, indirectly indicating a comparatively weaker interaction between starch and $Cl^-$. In other words, the lowest interaction between starch and $Zn^{2+}$ could endow an easy dynamic adsorption/desorption process of $Zn^{2+}$ during charge−discharge cycles, thereby not significantly affecting the predominantly $Zn^{2+}$-driven mass transfer process of the Zn-I FBs systems.

Long-chain polymer-typed starch units were featured with more significant bonding energy with $I_5^-$ polyiodide species (−0.66 eV) compared to $I_3^-$ species (−0.51 eV) in Fig. 2f[42], which indicates the amylose configuration could induce strong interaction between the starch and the iodine species, and the starch could consequently form a more stable combination with the $I_5^-$ active species. Furthermore, according to electrostatic potential (ESP) mapping, the charge distributions revealed that the regions near the hydroxyl regions of starch have more negative ESP values (Supplementary Fig. 21), considering the strong electron-donating center sites to the chemical interactions[35–37]. Meanwhile, acting as a Lewis acid, iodine atoms can react readily with electron-rich molecules via a charge transfer mechanism to form strong chemisorption complexes[32,33]. Therefore, the hydroxyl networks could form stabilized IS colloids by strong chemisorption, verifying the colloidal chemistry with aggregation effect.

## Electrochemistry of the colloidal electrolyte-based Zn-IS FBs

The electrochemical performances of Zn-IS FBs were evaluated to assess the effects of colloidal electrolytes on enhancing their cycling performance based on $2 \times 2$ cm$^2$ cells (Supplementary Fig. 22). As shown in Fig. 3a and Supplementary Fig. 23, the flow-mode Zn-IS FBs using the PP membrane (2 ml of 2 M $ZnI_2$ with 1 M starch || 8 ml of 2 M $ZnCl_2$ at 30 mAh) exhibited superior rate cycling and performance with high CE, VE and EE of 94−91−86, 96−85−83%, 98−81−78%, 98.5−75−74%, and 98.6−70−70% at 7.5, 15, 22.5, 30 and 37.5 mA cm$^{-2}$, respectively. Specifically, the CE of Zn-IS FBs gradually improved along with increased current density and can retain at the high value of over 98%, indicating the limited permeability of polyiodide active materials in the Zn-IS FBs during fast charging. Moreover, after switching the current density back to 7.5 mA cm$^{-2}$, the EE of Zn-IS FBs restored to 86%, indicating the excellent reversibility of colloidal starch-based cells at various rates. The corresponding voltage profiles of Zn-IS FBs using PP membrane were revealed in Fig. 3b, which demonstrated the relatively low voltage polarization in Zn-IS FBs due to the high ionic conductivity of porous PP membrane and the starch-based colloidal electrolytes. Furthermore, the result of chronoamperograms (CA, Supplementary Fig. 24) and cyclic voltammetry (CV, Supplementary Fig. 25) demonstrated that the colloidal starch in electrolytes can prevent the iodide/polyiodides from covering the electrode and thus facilitate the electrochemical reaction. In contrast, as displayed in Supplementary Fig. 26, although the Zn-I FBs without starch using conventional Nafion membrane (N117) show superior selectivity of $I_x^-$ active materials, N117-based FBs system exhibited inferior rate capability and larger polarizations on account of the low ionic conductivity and the significant internal resistance of the membrane. Particularly, PP membrane-based

Zn-IS FBs could deliver a high-power density of 41.58 mW cm$^{-2}$, demonstrating higher power compared to N117 membrane-based Zn-I FBs with a relatively low power density of 28.41 mW cm$^{-2}$ (Fig. 3c). Electrochemical impedance spectroscopy (EIS) result indirectly validated that Zn-IS FBs using PP membranes exhibit lower impedance compared to N117 membranes-based Zn-I FBs (Supplementary Fig. 27). Moreover, the internal resistance of the Zn-IS FBs was tested under the direct current (DC) mode to present the realistic resistance under the actual working conditions, which is different from the above-obtained EIS results under the alternating current (AC) testing condition (Note of Supplementary Fig. S28). As shown in Supplementary Fig. 28, based on the polarization voltage changes fitted at various DC, it could substantiate that the internal resistance of Zn-IS FBs system with starch using porous membranes is smaller compared to FBs without starch using N117 membranes. Therefore, the colloidal catholyte-enabled porous PP membranes could endow superior performance of Zn-I FBs compared with N117-based FBs. Notably, the N117 membranes-based Zn-I FBs and the developed Zn-IS FBs showed similar discharge capacity retention at different rates (Supplementary Fig. 29), indicating the suppressed cross-over of polyiodide active materials during cycles due to the size sieving effect by the colloidal starch.

The cycling performance of the Zn-IS FB at high current density and high-volumetric capacity was evaluated. Zn-IS FBs delivered a stable charge-discharge operation over 350 cycles at a high current density of 30 mA cm$^{-2}$ with high CE (98.5%), realizing the volumetric capacity of 6 Ah L$^{-1}_{catholyte}$ (Fig. 3d). Meanwhile, under a high volumetric capacity (33.5 Ah L$^{-1}_{catholyte}$) to achieve 50% utilization of the iodine, i.e., 50% SOC, this Zn-IS FBs flow system could demonstrate long cycling calendar life (over 250 cycles) with and high CE (~95%) at 22.5 mA cm$^{-2}$ (Fig. 3e), indicating the effective inhibition on the cross-over issue of iodine species. Under the same working condition, the PP membrane-based flow batteries in blank electrolytes without starch showed inferior CE at around 65% with severe capacity loss, lower discharging capacity as ~25 Ah L$^{-1}_{catholyte}$, and short cycle lifespan (~50 cycles) due to the severe cross-over and short-circuits (Supplementary Fig. 30). On the other side, although dense N117 ion-exchange membranes possess low $I_x^-$ permeation (Supplementary Fig. 31), N117-based Zn-I FBs failed in a short time of 150 h under high current density due to the notable by-products from severe side-reaction, which cause enlarged overpotential and the short-circuits of Zn dendrites to fluctuate the performance at the 42th cycles (Supplementary Fig. 32).

The side reactions during battery cycling are another critical issue that affects battery stability. Benefiting from stable colloid additives, aqueous colloid electrolytes as fast ion carriers can modulate the typical electrolyte system for improving reversible plating/stripping on Zn anode for high-performance Zn ion batteries[43,44]. The catholyte contains small-sized starch particles capable of migrating into the anolyte (Fig. 2c), which would form a colloidal electrolyte to a certain extent. Voltage profiles of Zn plating/stripping were studied based on asymmetric FBs (Zn foil@Carbon felt (CF) || PP membranes || CF, Supplementary Fig. 33) were evaluated for cyclic stability of anode. The Zn-based asymmetric FBs with colloidal starch (Supplementary Fig. 33b) exhibited a prolonged lifespan over 240 h (340 cycles) with a high CE of 99.01% at 30 mA cm$^{-2}$ under 10 mAh cm$^{-2}$, which is longer than that of the asymmetric FBs in blank electrolytes (over 150 h with low CE of 98.23%, Supplementary Fig. 33a) due to soft-short circuits caused by side reactions and zinc dendrites (the inset of Supplementary Fig. 33a). Moreover, the pH of blank electrolytes increased from 4.42 to 4.75 after 70 h and reached 5.13 after 140 h of operation in the inset of Supplementary Fig. 33a, which can be attributed to excessive side reactions in the blank electrolyte, resulting in the inferior reversibility of the Zn metal anode. In contrast, Zn-IS FBs with the starch-based electrolyte maintained a stable pH value throughout prolonged cycles in the Zn-IS operation.

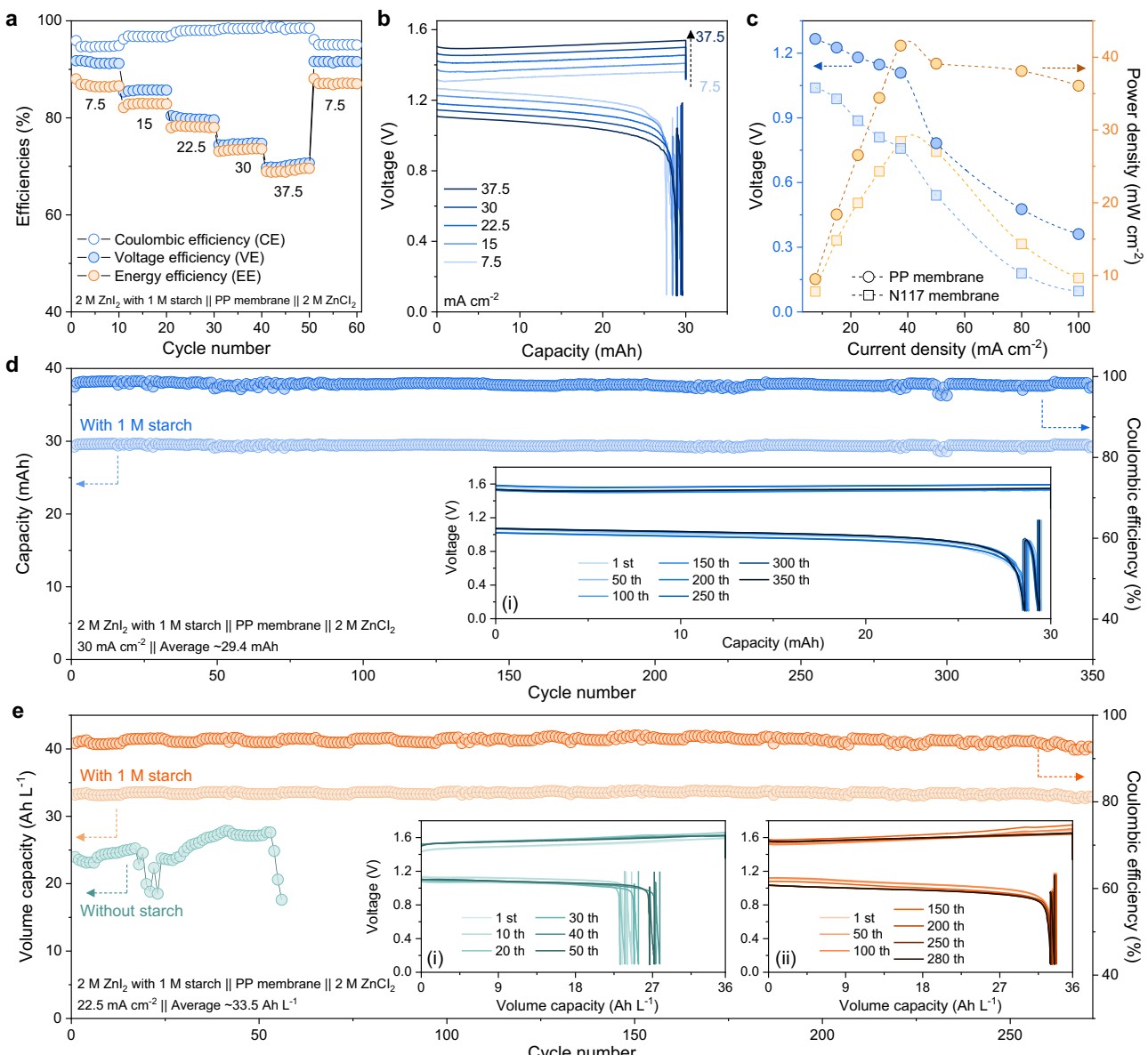

**Fig. 3 | Electrochemical performance of the Zn-IS FBs at 25 °C. a** CE, VE and EE of the Zn-IS FBs (2 ml of 2 M ZnI₂ with 1 M starch || PP membrane || 8 ml of 2 M ZnCl₂, 4 cm² membrane area) under 7.5, 15, 22.5, 30, 37.5 and 7.5 mA cm⁻². **b** Voltage profiles of the Zn-IS FBs under 7.5, 15, 22.5, 30 and 37.5 mA cm⁻². **c** The polarization of the Zn-IS FBs using different membranes (PP with 1 M starch & N117 without starch). **d** Cycling performances of Zn-I FBs flow-cell system at high current density (30 mA cm⁻²). **e** Cycling performances of Zn-IS FBs flow-cell system with/without starch at high volume capacity (50% SOC, 36 Ah L⁻¹) under a current density of 22.5 mA cm⁻². The inset in (**d**, **e**) are the corresponding voltage profiles.

As shown in Supplementary Fig. 34, SEM images of the CF at the Zn anode side of the discharge state exhibited irregular and mossy morphology with a rugged surface, which could be ascribed to the hydrogen evolution reactions to increase the pH of ZnCl₂ anolytes and produce the Zn₅(OH)₈Cl₂•H₂O and ZnO as by-products, as verified by the X-ray powder diffraction (XRD) results (Supplementary Fig. 35)[12,45]. In contrast, a smooth and cleaning surface was observed on the CF of the anode after the same cycling (Supplementary Fig. 36). The results could be attributed to the ultrasmall-sized colloidal starch that could cross the membrane to the anolyte and consequently stabilize the pH of the anolyte, hence endowing improved reversibility of the Zn anode. In particular, SEM images and XRD patterns of the PP membrane after cycles in starch-based colloidal electrolytes exhibited a cleaner surface compared to the membrane after cycles in blank electrolytes (Supplementary Figs. 36 and 37). Moreover, N117-based Zn-I FBs in blank electrolytes also showed severe side reactions at the Zn anode side

with dead dendrites and significant by-products (Supplementary Figs. 38–40). Therefore, starch-based colloidal chemistry can endow higher working currents and higher energy for the iodine cathode side, meanwhile promoting cycling stability for the Zn anode side and achieving improved performance for Zn-IS FBs systems.

To demonstrate the potential application of the starch-based colloidal electrolytes for the outdoor flow battery systems, the electrochemical performance of Zn-IS FBs was characterized at elevated temperatures (both the reservoirs and cells were kept at 50 °C, Supplementary Fig. 41). For the Iₓ⁻ permeability under high temperature of 50 °C (Supplementary Figs. 42 and 43), the colloidal starch could strongly confine the polyiodides by forming a colloidal aggregation featuring low Iₓ⁻ permeability to impede the cross-over issue even at a severe condition of high temperature. As shown in Fig. 4a, b and Supplementary Fig. 44, the Zn-IS FBs using the PP membrane (2 ml of 2 M ZnI₂ with 1 M starch || 8 ml of 2 M ZnCl₂ at 30 mAh) at high

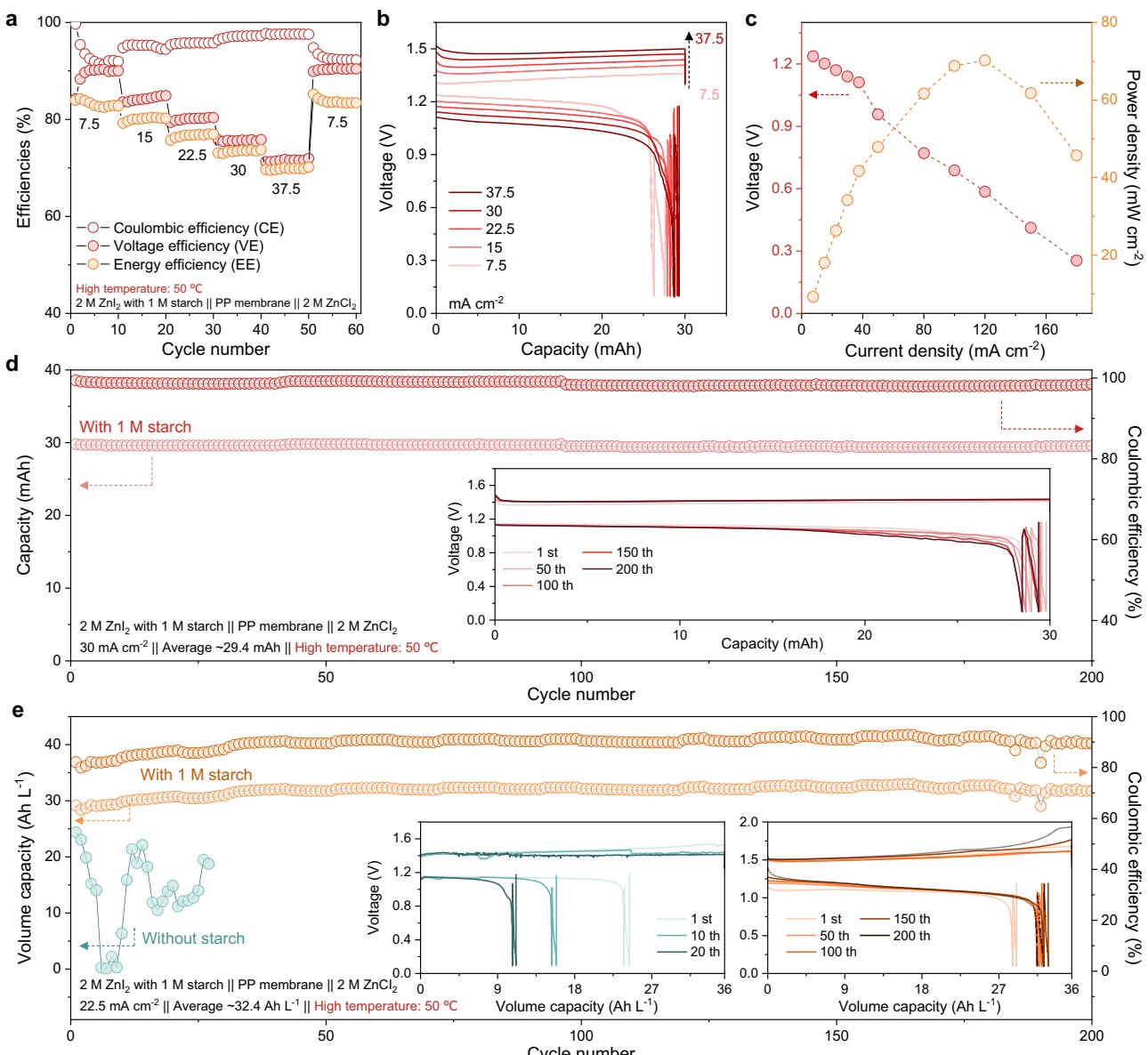

**Fig. 4 | Electrochemical performance of the Zn-IS FBs at 50 °C. a** CE, VE and EE of the Zn-IS FBs (2 ml of 2 M ZnI₂ with 1 M starch || PP membrane || 8 ml of 2 M ZnCl₂, 4 cm² membrane area) under 7.5, 15, 22.5, 30, 37.5 and 7.5 mA cm⁻² at high temperature (50 °C). **b** Voltage profiles of the Zn-IS FBs under 77.5, 15, 22.5, 30, and 37.5 mA cm⁻². **c** The polarization of the Zn-IS FBs using PP membranes at high temperature (50 °C). **d** Cycling performances of Zn-IS FBs flow-cell system at high current density (30 mA cm⁻²) and high temperature (50 °C). **e** Cycling performances of Zn-IS FBs flow-cell system with/without starch at high volume capacity (35.5 Ah L⁻¹) and high temperature (50 °C) under a current density of 22.5 mA cm⁻². The inset in (**d, e**) are the corresponding voltage profiles.

temperatures (50 °C) exhibited stable and reversible cycling ability with high CE, VE, and EE of 92–90–84%, 95–84–81%, 96–80–77%, 97–75–74%, 98–71–70% and 93–91–84% at 7.5, 15, 22.5, 30, 37.5 and back to 7.5 mA cm⁻², respectively. Notably, high CE showed in the initial cycles of the battery and then slight decreased as the number of cycles, which can be attributed to the oxidation of I⁻ by air with high-temperature conditions at initial flowing process, acting as the active species to supply the discharging capacity[46]. In addition, it can be observed that the volume capacity increased as the cycle number increased. which was due to undissolved Zn at high temperature (Supplementary Fig. 45)[47,48]. Those Zn can be further dissolved and contribute supplementary Zn metal for subsequent cycles to improve the CE. Moreover, the Zn-IS FBs still delivered a high power of 70.12 mW cm⁻² due to the smaller polarization with the high ionic conductivity at high temperatures (Fig. 4c). The Zn-IS FBs could accommodate a stable cycling operation over 200 cycles with a high

CE of 98.9% at a high current density of 30 mA cm⁻² charging mode under high temperature at 50 °C (Fig. 4d). In addition, when operating at a high-volume capacity of 35.5 Ah L⁻¹, the Zn-IS FBs based on colloidal chemistry exhibits superior cycling stability for 200 cycles with considerable CE (~91%) and discharging capacity of 32.4 Ah L⁻¹ at 50 °C (Fig. 4e). The cycling and charging/discharging profiles of the Zn-I FBs without starch colloids in the catholyte are shown in Supplementary Fig. 46. In stark contrast, it exhibited unstable cycling with fluctuating and low CE (below 40%), and a low volumetric capacity (~18 Ah L⁻¹_catholyte) at the same charged capacity and temperature. The Zn-IS FBs based on colloidal catholyte revealed notable higher CE and superior cycling stability. Such outstanding performance could be attributed to the above-discussed effective suppressions of the cross-over issue by the IS colloids by the size-sieving effect at the cathode side and the hydrogen evolution and Zn dendrite issues at the anode side.

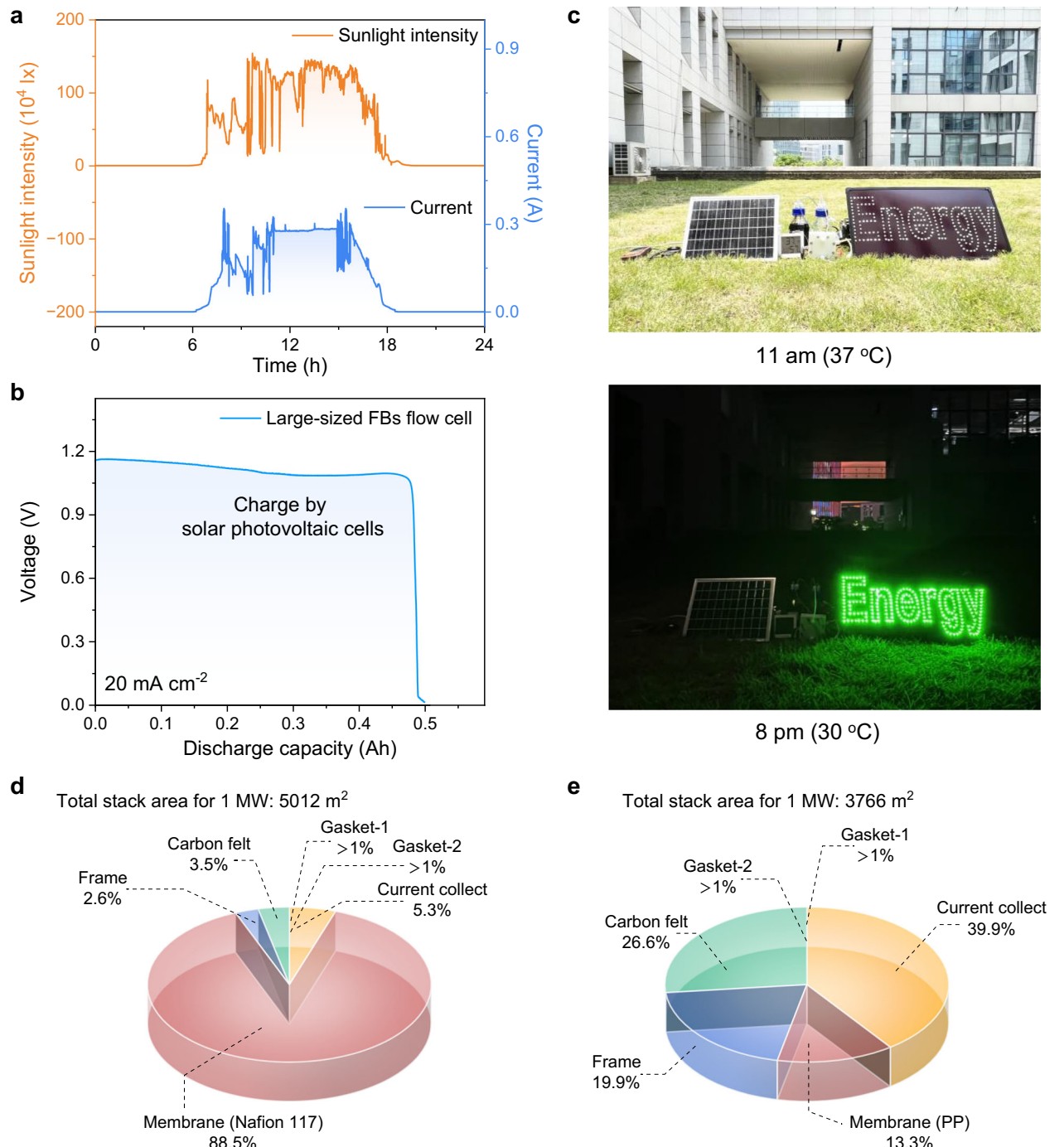

**Fig. 5 | Demonstration of the Zn-IS FBs for its practical application and the installed cost of 1 MW Zn-I FBs. a** Data of sunlight intensity (top) and charging current density (top) from the renewable energy storage demonstration system based on a large-sized Zn-IS FBs flow cell (250 ml of 1 M $ZnI_2$ with 1 M starch || PP membrane || 250 ml of 1 M $ZnCl_2$, 5 × 5 $cm^2$ membrane area) for one day. **b** Discharge voltage profiles of large-sized Zn-IS FBs flow cell after charging one day by solar photovoltaic cells at 20 mA $cm^{-2}$. **c** Solar-powered battery energy storage systems at day and night. The demonstrated solar-powered energy storage system is based on the Zn-IS FBs flow module as the energy storage device, a photovoltaic cell panel as a power source (rated at 12 W), and an LED display (280 green LED bulbs, rated at 16.8 W) serving as an electrical load. **d, e** Cost analysis based on a 1 MW Zn-I FBs flow stack based on Nafion 117 and PP membranes.

## Practical applications and cost analysis

The superior performance of the Zn-IS FBs motivated us to investigate further its potential to store renewable energy from solar sources. The scale-up Zn-IS FBs with enlarged areas of the membrane (25 $cm^2$ cell sizes, Supplementary Fig. 47) were integrated with solar photovoltaic panels. Furthermore, the data of solar intensity and corresponding solar-to-current conversion was detected through a controller for one day (Supplementary Fig. 48). As displayed in Fig. 5a, the output

currents of the solar cell stack, i.e., the charging currents for the Zn-IS FBs, began to rise after sunrise at 6:00 a.m. It approached 0.3 A and remained from approximately 12:00 to 15:00 p.m. and decreased after the sunset (18:30 p.m.). The fluctuations in the charging current could be attributed to the instability and the intermittency of the solar intensity. Overall, the energy conversion process was finished by converting the solar energy into electrochemical energy in the Zn-IS FBs. Then, the Zn-IS FBs were switched to the discharging mode, which

could deliver a capacity of 0.496 Ah and an energy of 0.535 Wh at 20 mA cm$^{-2}$, revealing the practical application of Zn-IS FBs as a power supply (Fig. 5b). Impressively, the Zn-IS FBs were demonstrated to power the logo "Energy" composed of 280 green LED bulbs, as shown in Fig. 5c. Note that ground temperatures could reach nearly 45.8 °C in the noonday sun at 2:00 p.m. (Supplementary Fig. 49 was recorded in South China). The successful integration of the scale-up Zn-IS FBs battery module with the photovoltaic cell panel demonstrated their high adaptability as large-scale energy storage systems in future smart grids.

The installed cost of the flow cell is one of the most critical factors to determine the commercialization potential for the demonstrated system. Here, we economically calculated the installed cost to construct a 1-MW zinc-iodine flow battery stack based on the Nafion 117 membrane and PP membrane with colloidal electrolytes, as shown in Fig. 5d, e. Based on the long-term cycling performance of the Zn-I FBs with the typical N117-based FBs of 19.95 mW cm$^{-2}$ and the as-developed PP-based FBs of 26.55 mW cm$^{-2}$ (Fig. 3c) at 22.5 mA cm$^{-2}$, the required stack membrane area for the simulated output 1-MW-system is 5012 m$^2$ of N117 membrane and 3766 m$^2$ of PP membrane systems. The detailed cost and the amount of components for the 1-MW flow cell stack are provided in Supplementary Table 1. Specifically, the Nafion membrane cost accounted for the highest proportion (88.49%, $500 per m$^2$), leading to the high installation cost of the full cell as $1,970,000.00 for the 1-MW cell. By contrast, by replacing the Nafion 117 membrane with the PP membrane, the cost of the 1-MW flow stack dramatically decreased 90.02% in the installed cost as $280,000.0, i.e., 14.3 times lower in cost. Such difference could be attributed to the membrane cost accounting for the lower proportion (13.32%, $10 per m$^2$) for the Zn-IS FBs, indicating that the reduced membrane cost could significantly reduce the installed cost of the flow battery in the practical application. As shown in Supplementary Table 2, we further analyzed the cost of 1-MW-system Zn-I FBs for a long-duration output of 1 MWh. Benefiting from the excellent output power in PP membranes-based Zn-IS FBs systems, the simulated Zn-IS stack (total $636,013.00) presented a lower price in a 1-MWh-energy storage system than N117-based Zn-I FBs stack (total $3,184,189.00). Therefore, it can be foreseen that further optimization of the colloidal chemistry-based flow battery components can advance a new arena of next-generation zinc-based flow batteries with power cost-effectiveness and remarkable energy density for future grid-scale energy storage applications in hot climates.

## Discussion

In summary, we developed the starch-containing colloidal chemistry for improved polyiodide selectivity based on the size-sieving effect, which enabled the low-cost porous PP membranes as LPPM with superior ion conductivity suitable for Zn-I FBs. It could allow the high working currents, high power and stable cycling of the Zn-IS FBs based on cheap PP membranes. As a result, the Zn-IS FBs based on colloidal catholyte could accommodate the high current density of 37.5 mA cm$^{-2}$, a power density of 42 mW cm$^{-2}$, high CE of 98.5% and stable cycling over 350 cycles. The as-developed colloidal Zn-IS FBs systems could deliver stable cycling over 250 cycles at a high volumetric capacity of 32.4 Ah L$^{-1}$ (50% SOC) even under a high temperature of 50 °C benefiting from the effective size-sieving of the starch-iodine complex based on the strong chemisorption. Moreover, the applied starch colloids could improve the Zn anode reversibility and the cycling stability of the Zn-IS FBs. According to a cost-simulated analysis based on a 1-MW-flow cell, the installation cost of the PP membrane-based stack dramatically decreased 10.1 times that based on the Nafion membrane. Furthermore, the scaled-up flow battery module exhibited the potential to combine with photovoltaic solar packs as integrated renewable energy storage systems. This work would serve as a model system to exploit colloidal electrolyte

chemistries to develop LPPM-based flow batteries with low-cost, high-power and high-temperature adaptability for large-scale energy storage.

## Methods

### Materials

All chemicals were used as received. Zinc iodide (ZnI$_2$, ≥99.99%), zinc chloride (ZnCl$_2$, ≥99.99%), starch soluble ((C$_6$H$_{10}$O$_5$)$_n$, ≥99%), potassium iodide (KI, ≥99%), potassium hydroxide (KOH, ≥8 %), sulfuric acid (H$_2$SO$_4$, 95–98%), hydrogen peroxide (H$_2$O$_2$, 30 wt% in H$_2$O), iodine (I$_2$, ≥99%) were received from Sigma-Aldrich. Graphite felt (3.0 mm, carbon ≥99 %, bulk density 0.12-0.14 g cm$^{-2}$) was obtained from Yi Deshang Carbon Technology. Nafion membrane (N117, Dupont) was received from Shanghai Hesen Electric. Polypropylene (PP) membrane (Celgard 2325) was received from Suzhou Sinoro Technology. Ti mesh (99.9%, 100 mesh) was obtained from Kangwei Metal. Zn foil (200 μm, 99.99%) was purchased from Chenshuo Metal.

### Preparation of Zn$^{2+}$ typed ion exchange membrane

The Zn$^{2+}$-type Nafion membranes (N117 from Dupont, USA, active area of 9 cm$^2$) were pretreated prior to use with the following steps. Firstly, the Nafion membrane was soaked in 3 wt% H$_2$O$_2$ aqueous solutions at 80 °C for 1 h. It was rinsed with deionized (DI) water to remove the remaining H$_2$O$_2$. Secondly, the Nafion membrane was soaked in 0.5 M H$_2$SO$_4$ at 80 °C for 1 h and rinsed with DI water until the pH in the effluent was near 7. Finally, it was soaked in 1 M ZnCl$_2$ aqueous solution (pH was adjusted to 1 by using HCl) at 60 °C for 3 h to convert from the H$^+$-type to the Zn$^{2+}$-type. Then the membrane was rinsed with DI water and stored in DI water.

### Polyiodides permeability

The permeability of KI$_3$ under PP membranes was determined by the UV-visible spectra, which tested the permeate side in H-cell tests in Supplementary Fig. S3. The feed side contained 2 M KI$_3$, and the permeate side contained DI water. This H-cell featured a transport channel with circular symmetry and separated by PP membranes. During the experiments, it was postulated that the change in KI3 concentration with the feed solution was negligible when permeation side displayed the low concentration and the flux of KI$_3$ by PP membrane remained constant. The permeation can be calculated as[28]:

$$V_B \frac{dC_B(t)}{dt} = \frac{AP}{L} \left[ C_A - C_B(t) \right] \tag{1}$$

$$\ln\left( 1 - \frac{C_B(t)}{C_A} \right) = -\frac{AP}{LV_B}[t - t_0] \tag{2}$$

where $V$ (ml) represents the volume of the permeate solution; $C_A$ and $C_B(t)$ (mol L$^{-1}$) represent the concentrations of KI$_3$ solution in the feed and permeate side, respectively; $A$ (cm$^2$) and $L$ (cm$^2$) represent the area and thickness of membranes, respectively; $P$ (cm$^2$ min$^{-1}$) represents the membranes permeability; $t$ (min) and $t_0$ (min) represent the time and the time lag, respectively. The permeability $P$ can be determined from the slope fitted by $-\ln(1 - C_B(t)/C_A)$ and $t$.

### Electrochemical characterization

Catholyte composed of 2 M ZnI$_2$ was dissolved in deionized water. Anolyte was prepared with 2 M ZnCl$_2$ in deionized water. The zinc-iodine flow batteries (Zn-I FBs) cell assembly configuration: briefly, polytetrafluoroethylene (PTFE) frames served as the flow channel to fix the position of the pretreated three-dimensional electrodes with a geometric area of 4.0 cm$^2$ (2 × 2 cm$^2$) or 25 cm$^2$ (5 × 5 cm$^2$) and thickness of 2.0 mm (Supplementary Fig. 9). Carbon felt (CF) was utilized as both the positive and negative electrode. To make a flow-mode cell, a

peristaltic pump (Chuang Rui Precision Pump) was employed to power the circulation of the electrolyte flow through the electrodes.

CV and CA tests were conducted on an electrochemical workstation (CHI 760E) with a three-electrode system where glassy carbon electrode (6 mm in diameter) was used as working electrode, platinum plate as counter electrode and saturated calomel electrode (SCE) as the reference electrode in the electrolytes. The working potential was operated with stirring and tested from −0.4 to 1.2 V vs. SCE. The CA test was conducted at a constant potential of 1.2 V vs. SCE.

EIS measurements were carried out on a CHI electrochemical testing unit (760E). The sinusoidal voltage oscillations of 10 mV amplitude at the OCV of the cells were collected after charging to 50% SOC. The oscillation frequencies ranged from 1000 kHz to 100 mHz with three repetitions for every test. Internal resistance of FBs was fitted by the polarization voltage under different current density (10–40 mA cm$^{-2}$), wherein FBs after charging to 50% SOC. was charging and discharging for 1 min at different current density.

**The galvanostatic performance of the Zn-I FBs.** The batteries were testing on a battery system (LAND, CT2001A) at ambient temperatures (20–25 °C) or reacted in an incubator at high temperatures (50 °C). The current densities were set in the range 7.5 to 37.5 mA cm$^{-2}$. The charging mode was limited by the constant capacity (30 mAh or 71 mAh) and discharging mode was limited by the cut-off voltage of 0.1 V. The theoretical capacity was determined based on the catholyte comprising the iodide part, which constituted the limiting capacity factor for the full cell. Catholyte adjusted to final volume of 2 mL to afford a 4 M electron concentration (the capacity of 1 M charge density is 26.8 mAh mL$^{-1}$, total 214.4 mAh).

**Zn-based asymmetrical flow batteries test.** The battery was assembled with carbon felts (2 cm × 2 cm × 0.46 cm) used as anodes, Zn metal (2 cm × 2 cm × 0.1 cm) and blank CF (2 cm × 2 cm × 0.46 cm) used as cathode, Celgard 3501 membrane and 20 mL of 2 M ZnCl$_2$ with/without 1 M starch used as electrolyte to flow past the two electrodes. The test was examined by plating onto carbon felts at 30 mA cm$^{-2}$ and 10 mAh cm$^{-2}$, followed by stripping Zn from these substrates to a cut-off voltage (-0.5 V) by battery testing system.

**The renewable solar-energy storage system.** The Zn-I FBs cell pack (25 cm$^2$, 5.0 × 5.0 cm$^2$) was integrated into a wind and photovoltaic power generating system. During the operation of the Zn-I FBs, no battery management system was used to control each individual Zn-I FBs to demonstrate the working flexibility of the Zn-I FBs under fluctuating charge voltage. The output of the photovoltaic power cell was connected to the input of the ultralow-power DC–DC boost converter bq25504 EVM with constant voltage of 2 V and the Zn-I FBs cell pack was connected toed to the output of the converter for charging. The solar light intensity was collected and measured by the sunlight detection Sanliang-PP730. The output current was collected by CHI measurement.

### Characterizations

The crystal structure was studied by X-ray diffraction (XRD, X'Pert Pro MPD, Philips, Holland) using Cu Kα as the radiation source under 40 kV and 40 mA. Morphologies were probed by scanning electron microscopy ((SEM, FEI Quanta 450 FEG SEM). X-ray photoelectron spectroscopy (XPS) spectra were recorded on a photoelectron spectrometer (ESCALAB 250, Thermo Scientific, America), where the binding energy (BE) of the elements was calibrated by the BE of C 1s (284.60 eV). The modulus mapping was measured by atomic force microscope (Bruker, DIMENSION ICON) and conducted in the quantitative nano-mechanics mode (QNM). Raman measurement (Dxr-2xi, Thermo Scientific, America) was performed with in situ homemade cells to observe the O-H stretching peak. The pH meter (Brand: REX, Model: PHBJ-261L)) was used to monitor pH changes.

### Density functional theory (DFT) calculation

Electrostatic potential (ESP) mappings were carried out with the Gaussian 09 W software package to gain structural information on the abovementioned molecules. Geometrical optimization adopted the B3LYP method with 6-31 + G(d,p) basis sets. Based on the optimized structure of molecules, ESP analysis on van der Waals surface was done to deduce the possible soluble starch functional sites using the Multiwfn 3.3.8 software package in the Gaussian 09 W software package[49].

The structures of Zn$^{2+}$, Cl$^-$, I$^-$, I$_3^-$ or I$_5^-$, soluble starch, and their complex (starch@Zn$^{2+}$, starch@Cl$^-$, starch@I$^-$, starch@I$_3^-$ or starch@I$_5^-$) were first optimized by using the density functional theory (DFT) at the B3LYP/def2-TZVP level[50]. All geometry optimizations, including the implicit solvent model with SMD were performed using the DFT-D3 method in ORCA[51,52]. Then, the single-point energies of complexes were done at the same level after the previous optimization, which considered basis set superposition error (BSSE). The harmonic frequency calculations were carried out at the same level of theory to help verify that all structures have no imaginary frequency.

The binding energy of the configuration ($E_{bind}$) was calculated by the following equation:

$$E_{bind} = E_{AB} - (E_A + E_B) \tag{3}$$

where $E_A$, $E_B$, and $E_{AB}$, respectively, represent the energies of A (Zn$^{2+}$, Cl$^-$, I$^-$, I$_3^-$ or I$_5^-$) and B (soluble starch) and the complex energy, a negative value of $E_{bind}$ indicates that the process is an exothermic reaction and a high negative value corresponds to a stronger interaction, which indicates more heat release and a more stable product.

### Reporting summary

Further information on research design is available in the Nature Portfolio Reporting Summary linked to this article.

## Data availability

All data that support the findings of this study are presented in the manuscript and Supplementary Information or are available from the corresponding author upon reasonable request. Source data are provided with this paper.

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

## Acknowledgements

This research was supported by GRF at CityU 11304921 and Guangdong Basic and Applied Basic Research Foundation under Project 2022B1515120019.

## Author contributions

C.Z. conceived the project. G.L. supervised the research. Z.W. and Z.H. prepared the materials. Z.W., Z.H., Y.W., and Y.Y. conducted the characterization and analyzed the data. Z.W., S.W. and T.H. performed density functional theory calculation. Z.W., G.L., and C.Z. wrote the paper, and all authors engaged in discussions related to the manuscript.

## Competing interests

The authors declare no competing interests.
