## [Peer Review File · Nature Communications]

Starch-mediated Colloidal Chemistry for Highly Reversible Zinc-based Polyiodide Redox Flow BatteriesREVIEWER COMMENTS

Reviewer #1 (Remarks to the Author):

In this manuscript, the authors created the starch-containing colloidal chemistry for improving the ion (polyiodide) selectivity based on the size-sieving effect and demonstrated the performance for the Zn-IS FBs application. The authors investigated the effect of colloidal aggregation on viscosity, permeability and conductivity as essential ex-situ parameters to determine the overall performance of FBs. In addition, DFT calculations were performed to support a stable combination of colloidal aggregation between starch and polyiodide. Overall, the prepared Zn-IS FBs exhibit high coulombic efficiency and superior cycling stability even at a high temperature of 50 °C. Finally, the authors conducted a cost simulation analysis and concluded that the installation cost of the PP membrane based stack was dramatically reduced by 15 times compared to the Nafion membrane based stack.

Notes.

1. The authors should include a more detailed discussion and explanation of how the starch-polyiodide complex affects the electrochemical reaction at the electrode during the charging and discharging process. The reviewer is concerned that the current approach, while unique and effective for preventing polyiodide molecules, will affect the electrochemical reaction at the electrode resulting in decreased efficiency.
2. The authors should include a more detailed discussion and explanation of the interaction between starch and other charge carrier molecules rather than polyiodide anions. The reviewer wonders if the strong chemisorption to starch is selective for polyiodide molecules or not. The reviewer is concerned that starch-charge carrier complex could inhibit charge carrier transport rate resulting in high resistance.
3. On page 5, Fig. 2b, starch concentration from 0 M to 1 M, the viscosity looks no different, but the ionic conductivity gradually decreases. Authors should explain why ionic conductivity and viscosity trend are different.
4. In the IS catholyte permeation test, please explain why the authors use DI water as blank solution. The reviewer is concerned about unbalanced osmotic pressure and ionic strength during the permeation test.
5. On page 6, 7, the authors should correct a number of figure numbering errors.
6. On page 8, "(d) Cycling performance of the Zn-I FBs flow cell system at high power density (30 mA cm²)". Perhaps they meant current density?
7. The authors should provide the permeability, CE, VE, and EE data of the N117 membrane and please compare them to PP with 1 M starch.
8. On page 10, Fig. 4a, at a current density of 7.5 mA cm⁻², CE decreases as the number of cycles increases from 0 to 6. The authors should explain why it is not stable and decreases.
9. On page 10, "The results could be attributed to the ultras-small colloidal starch, which could cross the membrane to the anolyte and consequently stabilize the pH of the anolyte, thus

conferring improved reversibility of the Zn anode." The authors should provide more detailed mechanisms or references for this statement.

10. The authors should provide permeation test data for 50 °C conditions.

11. On page 10, Fig. 4e, at cycle number from 0 to 50, the authors should explain why the volume capacity increases as the cycle number increases. It looks different from the electrochemical performance at 25 °C.

Reviewer #2 (Remarks to the Author):

The authors utilized starch to form complexes with iodine ions, effectively increasing the molecular size. This approach was aimed at preventing crossover and enabling the use of more cost-effective hydrocarbon membranes in the Zn-I flow battery system. Their results demonstrated stable cycling performance when starch was used as an additive. However, there are several areas where the manuscript could be improved:

1. While the authors explored the impact of starch on the physical properties of the iodine electrolyte, they did not conduct an electrochemical study to investigate how starch affects the crucial thermodynamic redox behavior and kinetics of the iodine/iodide couple. This is an essential aspect that should be addressed.

2. The Zn-I system is known for its high energy density due to the high solubility of ZnI salt. The authors only tested up to a 2M concentration of ZnI. It would be valuable to explore what happens when the concentration is increased to its solubility limit and whether starch remains effective. A more comprehensive study in this regard is recommended.

3. Figure 3d shows a significant difference in the capacity of the cell with and without starch. This difference needs a clear explanation. It should be clarified whether these tests were conducted at the same concentration, and if not, the reasons for the difference should be provided.

3. The inclusion of a cost analysis is commendable, but the study would benefit from a more comprehensive assessment of the overall system cost, including the cost of chemicals. This additional information would provide readers with a more precise understanding and facilitate comparisons with other systems.

In conclusion, this study is intriguing, but it would greatly benefit from revisions in the mentioned areas. I encourage the authors to make these improvements and resubmit their manuscript for further consideration.

Reviewer #3 (Remarks to the Author):

Development of low-cost and efficient energy storage technologies are of key importance for realizing the energy transition towards a clean model based on renewable sources. Among the various emerging technologies, aqueous zinc-iodine flow batteries (Zn-I FBs) holds great promising due to its low cost and the independent scalability of power and energy. However, there are several issues that prevent practical implementation of this promising technology. In this context, the authors report a strategy to avoid the use of expensive ion-selective

membrane while improving the energy efficiency. They propose the replacement of ion-selective membrane by porous separators. While this strategy has been long desired, porous separator leads to cross-contamination of positive and negative species, which results in short lifespan. The proposed strategy is based on the addition of starch which results in a complexation with polyiodide species. This complexation increases the size of the active species, so that size exclusion porous separators are able to prevent the cross-contamination.

The hypothesis is sound, and the findings reported in the manuscript seem to support the hypothesis. The demonstrated power density and specially the lifespan are very relevant in this field. Overall, this work could become suitable for publication in Nature Communication as long as the authors are able to address four major issues:

- The internal resistance when ion-selective membrane is used is claimed to be drastically increase with respect to the proposed strategy. As a result, the power density achieved using N117 Nafion membrane was only of 28 mW cm⁻² against 70 mW cm⁻² for 1 M starch. The EIS measurements are used to explain the poor power density. However, the difference for Nafion and PP separators is only 0.16 Ohm cm⁻². At 40 mA cm⁻², the voltage drops induced by such a difference would be 4.5 mV. Thus, charge transfer resistance and diffusion must be the origin. However, the low frequencies resistance including all parameters is even lower than 0.16 Ohm cm⁻², which is due to the presence of starch. Therefore, the differences in power density are not supported by the results. This point must be explained properly.
- Cross over of water due to osmotic issues. Having asymmetric electrolyte will have an impact in the viscosity and osmotic properties. The transfer of water between the two compartments is likely to occur in the long run. This point needs to be discussed as it may hinder implementation. How do the authors propose to overcome this relevant issue?
- Coulombic efficiency. The value of Coulombic efficiency is 98 %. Considering the low permeability value, and the relatively short length of one cycle, cross over of polyiodide does not seem to be the source of the Coulombic inefficiency. It is necessary to estimate the impact of the cross-contamination in one cycle. If the inefficiency is coming from the negative side, how can the systems operate for having such a large amount of HER and pH change? This point requires proper explanations.
- Cost analysis. The power value used for cost analysis are peak power values, which were not shown to be feasible under continuous operation. The comparison here was conducted at 22.5 mA cm⁻² (36 Ah L⁻¹), so that the cost analysis should be conducted at the power realized at this current density. While fair price is used for Nafion membrane, the cost of bipolar plates and graphite felt is underestimated since both are > 50 USD m⁻² in most of the reports. As a result, the cost of the proposed systems at the demonstrated conditions will be significantly higher.

Responses to Reviewer's Comments

Reviewer #1:

In this manuscript, the authors created the starch-containing colloidal chemistry for improving the ion (polyiodide) selectivity based on the size-sieving effect and demonstrated the performance for the Zn-IS FBs application. The authors investigated the effect of colloidal aggregation on viscosity, permeability and conductivity as essential ex-situ parameters to determine the overall performance of FBs. In addition, DFT calculations were performed to support a stable combination of colloidal aggregation between starch and polyiodide. Overall, the prepared Zn-IS FBs exhibit high coulombic efficiency and superior cycling stability even at a high temperature of 50 °C. Finally, the authors conducted a cost simulation analysis and concluded that the installation cost of the PP membrane-based stack was dramatically reduced by 15 times compared to the Nafion membrane-based stack.

General Response: Thanks for your professional evaluation of our work. We appreciate your positive assessments. We have provided more explanations and experiment results to address your comments.

Comment 1: *The authors should include a more detailed discussion and explanation of how the starch-polyiodide complex affects the electrochemical reaction at the electrode during the charging and discharging process. The reviewer is concerned that the current approach, while unique and effective for preventing polyiodide molecules, will affect the electrochemical reaction at the electrode resulting in decreased efficiency.*

Response: Thanks for your constructive comments. Following your suggestions, we applied chronoamperometry (CA) and cyclic voltammetry (CV) to evaluate the starch-polyiodide complex's effect on the electrochemical reaction. We discussed the results as follows.

a) To explore the effect of starch on the reaction kinetics of iodine, we conducted a chronoamperometry (CA) test using the KI electrolyte with/without starch at a constant potential of 1.2 V (**Fig. R1**). For blank KI solutions, it is observed that the current experiences a sharp decrease in the

first two seconds, followed by a steady-state value of approximately 1.01 mA. The decline can be ascribed to the formation of solid I_2 that covers the electrode, passivating the reaction of the active material in the initial reaction (*J. Mater. Chem. A*, 2022, **10**, 14090). In contrast, for the KI electrolytes with starch, the current showed a much larger value of 7.29 mA and remained constant during the CA test, which indicated the starch could rapidly interact with products (I_2/I_3^-), dissolve into the electrolyte, and continuously uncover the fresh surface of the electrode, which favors the redox reaction.

Fig. R1 | Chronoamperograms in 0.1 M KI and 0.1 M KI with 1 M starch.

b) Then, the cyclic voltammetry (CV) curve of I_3^-/I^- redox reactions in potassium iodide (KI) electrolytes with/without starch in the potential range of -0.4 to 1.2 V (vs. SCE) was tested by the three-electrode mode. To simulate a real flow battery environment, the electrolyte was stirred to ensure sufficient dispersion of the oxidized I_3^- into the electrolyte (**Fig. R2a**). Due to excess iodide ions, the CV curves during the anodic scan did not show the complete oxidation peak (**Fig. R2 (b & c)**). Moreover, the Tafel plots were calculated by corresponding CV curves. As shown in **Fig. R2d**, the starch-electrode was characterized by lower Tafel slopes for both oxidation (160 mV dec^{-1}) and reduction (13 mV dec^{-1}) steps than those of blank electrolytes-based electrode (oxidation: 238 mV dec^{-1} , reduction: 160 mV dec^{-1}), proving the significantly enhanced reaction kinetics of iodide species in both charging and discharging processes in Zn-I FBs. Those results can be attributed to the blocked electrode by passivating solid I_2 in blank electrolytes while keeping the clean surface of the electrode with starch.

Fig. R2 | (a) The photograph of the reactor under flowing-mode conditions. Cyclic voltammetry of redox reactions in (b) 0.1 M KI and (c) 0.1 M KI with 1 M starch under flowing-mode conditions. (d) the corresponding Tafel plots in different electrolytes.

As demonstrated in the above analysis, the colloidal starch in electrolytes can facilitate the electrochemical reaction of iodide/polyiodides. Although it influences the mass transfer of active species due to the ionic conductivity, the colloidal starch contributes to a stable cathode with high electrochemical reactivity by avoiding iodide/polyiodide covering on the electrode surface. We are sorry for not providing these results in our original manuscript. According to your suggestion, we have provided them on **Page 9** in the revised manuscript and **Supplementary Fig. 24 & Fig. 25 (Page S26 & S27)**.

Comment 2: *The authors should include a more detailed discussion and explanation of the interaction between starch and other charge carrier molecules rather than polyiodide anions. The reviewer wonders if the strong chemisorption to starch is selective for polyiodide molecules or not. The reviewer is concerned that starch-charge carrier complex could inhibit charge carrier transport rate resulting in high resistance.*

Response: Thanks for your kind suggestions. We have analyzed the impact of starch on different charge carriers from the following three aspects (the corresponding ionic conductivity, ions transference number, and DFT), where this work involves Zn^{2+} , Cl^- , and I^- ions.

a) As shown in **Fig. R3**, the ion conductivity of different starch concentrations in the ZnCl_2 solution was consistently higher than in the ZnI_2 solution from 0.1 M to 3 M. This suggests that starch would have a stronger interaction with iodide I^- ions than with Cl^- ions, resulting in a substantial number of anions (I^-) being locked in the ZnI_2 solution, hence exhibiting the lower ion conductivity.

Fig. R3 | Ionic conductivity of 2 M ZnI₂ and 2 M ZnCl₂ electrolytes with different starch concentrations (0, 0.1, 0.2, 0.5, 1, 2, and 3 M, M = mol L⁻¹).

b) According to the ion transference number (*Nat Commun*, 2023, **14**, 760; *Nat Commun*, 2022, **13**, 4181) calculated from Electrochemical Impedance Spectroscopy (EIS) and Chronoamperometry (CA) in **Fig. R4 & Fig. R5**, the ZnI₂ solution with colloidal starch exhibited a higher cation (Zn²⁺) transference number ($t_+ = 0.651$) compared to the blank ZnI₂ solution ($t_+ = 0.484$), which can be attributed to the entrapment of anions (I⁻) by starch. Conversely, the cation transference number of the ZnCl₂ solution with colloidal starch was only a marginal increase compared to that of the blank ZnCl₂ solution, indirectly indicating a comparatively weaker interaction between starch and Cl⁻.

Fig. R4 | Chronoamperometry curves of Zn||Zn symmetrical cell with a static potential of 10 mV in (a) blank 2 M ZnI₂, (b) 2 M ZnI₂ with starch, (c) blank 2 M ZnCl₂ and (d) 2 M ZnCl₂ with starch. The corresponding EIS plots before and after polarization are shown in the inset.

Fig. R5 | Transference number of blank 2 M ZnI₂, (b) 2 M ZnI₂ with 1 M starch, (c) blank 2 M ZnCl₂ and (d) 2 M ZnCl₂ with 1 M starch.

c) DFT calculations (*Nat Commun*, 2023, **14**, 6526; *Phys. Rev.* 1965, **140**, A1133; *J. Phys. Chem. B*, 2009, **113**, 6378) were employed to explore further the interaction between colloidal starch and different charge carriers. As depicted in **Fig. R6 & Supplement Table 1**, the binding energy among different samples demonstrates the order of Starch@I⁻ ($E_b = -0.354$ eV) > Starch@Cl⁻ ($E_b = -0.119$) > Starch@Zn²⁺ ($E_b = -0.081$ eV). Therefore, the lowest interaction between starch and Zn²⁺ could endow an easily dynamic adsorption/desorption process of Zn²⁺ during charge-discharge cycles, thereby not significantly affecting the predominantly Zn²⁺-driven mass transfer process for Zn-I FBs systems.

Fig. R6 | The evolution of bonding energy of (a) I⁻, (b) Cl⁻ and (c) Zn²⁺ interacting with the soluble starch.

Supplementary Table 1 | The binding energy of the configuration between starch and different ions.

Sample	E_{A-Ions} (Ha)	$E_{B-Starch}$ (Ha)	E_{AB} (Ha)	E_{bind} (eV)

Starch@I⁻	-297.786240	-1297.925955	-1595.725219	-0.3544
Starch@Cl⁻	-353.780518	-1297.925955	-1651.71087	-0.11965
Starch@Zn²⁺	-1778.816150	-1297.925955	-3076.74511	-0.08177

In summary, as analyzed above, colloidal starch primarily interacted strongly with polyiodides while exerting a minor influence on other charge carriers. The interaction intensities with the ions are sequenced as $I_x^- > I^- > Cl^- > Zn^{2+}$. There was only a mild effect on the charge carrier transport rate and ionic conductivity due to the influence of the viscosity and limited anion. Therefore, the selectively interactive starch electrolyte can maintain a specific charge carrier transport, ensuring a favorable charge-discharge process in Zn-IS flow batteries. We are sorry for not providing these results about the impact of charge carrier transport by starch in our original manuscript. Per your suggestion, we have provided them on **Page 7** in the revised manuscript and **Supplementary Fig. 18 - 20 (Page S20 - S22)**.

Comment 3: *On page 5, Fig. 2b, starch concentration from 0 M to 1 M, the viscosity looks no different, but the ionic conductivity gradually decreases. Authors should explain why ionic conductivity and viscosity trend are different.*

Response: Thanks for the constructive comments from the reviewer. We believe this is an illusion caused by the inappropriate choice of the vertical axis in **Fig. 2b** (in the original manuscript). As shown in **Fig. R7**, when we magnified the viscosity range from 0 to 1 M starch, a noticeable increasing trend in viscosity became apparent. Specifically, the viscosity of the electrolytes was 0.71 (10^{-3} Pa S) at 0 M starch and 14.5 (10^{-3} Pa S) at 1 M starch, while the ionic conductivity (**Fig. R7**) showed a decrement from 100.1 ($mS\ cm^{-1}$) at 0 M to 43.7 ($mS\ cm^{-1}$) at 1 M. It can be observed that both viscosity and ionic conductivity exhibited a noticeable change, in which an increase in viscosity resulted in a reduction of ionic conductivity.

Fig. R7 | viscosity with different starch concentrations (0, 0.1, 0.2, 0.5, 1 M).

On the other hand, the factors affecting the ionic conductivity of the electrolytes are complex, encompassing the viscosity, salt concentration, temperature, ion characteristics (charge and size), solvent properties, and the types of ions present, collectively influencing the overall conductivity levels (*Nat. Mater.*, 2020, **19**, 1006; *Nat Energy*, 2019, **4**, 269; *J. Am. Chem. Soc.*, 1916, **38**, 2431). However, as elucidated in the above **Comment 2**, the decreased ionic conductivity can be due to the strong locking effect of starch on I⁻ anions and the increased viscosity to jointly retard the ion transport. Meanwhile, to improve readability for readers, we modified the range mode of ion conductivity (log scale), as shown in **Fig. R8**. It can be clearly observed that the ionic conductivity exhibited a significant change in magnitude only when the starch concentration reached 3 M. Therefore, a consistent trend existed between the ion conductivity and viscosity across different starch concentrations. The related discussion has been added to the revised manuscript (**Fig. 2b, Page 5**).

Fig. R8 | Viscosity, ionic conductivity, and I_x^- permeability of 2 M ZnI_2 electrolytes with different starch concentrations (0, 0.1, 0.2, 0.5, 1, 2, and 3 M).

Comment 4: *In the IS catholyte permeation test, please explain why the authors use DI water as a blank solution. The reviewer is concerned about unbalanced osmotic pressure and ionic strength during the permeation test.*

Response: Thanks for your constructive comments. As highlighted by your suggestions, the unbalanced osmotic pressure and ionic strength between the electrolytes on both sides during the permeation test in an H-cell configuration can result in significant permeation issues. Meanwhile, osmotic pressure is also influenced by factors such as the concentration of solutes, temperature, the nature of the solvent and solute, and the properties of the membrane (*J. Am. Chem. Soc.*, 1908, **5**, 668; *Nat Sustain*, 2020, **3**, 296). Emphasizing the inhibition of polyiodides cross-over by the colloidal starch, using single salt dialysis (DI water as the permeate side) represented a harsh testing condition, creating a significant osmotic pressure and ionic strength differential environment (*Nat Energy*, 2021, **6**, 517; *Nat. Mater.*, 2020, **19**, 195). Under such conditions, if the IS-based system exhibits a relatively low permeation rate, it indicates even lower permeation rates under other conditions. Hence, we believe this situation could better illustrate the pivotal role of starch in inhibiting the cross-over of polyiodides. We are sorry for not providing this information in our original manuscript, and it was discussed in **Supplementary Fig. 3 (Page S5)**.

Comment 5: *On page 6, 7, the authors should correct a number of figure numbering errors.*

Response: We are sorry for our oversight. We have revised them accordingly, as highlighted in yellow in our revised manuscript (**Pages 5 - 7**).

Comment 6: *On page 8, "(d) Cycling performance of the Zn-I FBs flow cell system at high power density (30 mA cm²)". Perhaps they meant current density?*

Response: We are sorry for our oversight and we have corrected it accordingly, as highlighted in yellow in our revised manuscript (**Page 8**).

Comment 7: The authors should provide the permeability, CE, VE, and EE data of the N117 membrane and please compare them to PP with 1 M starch.

Response: We appreciate your helpful and valuable suggestions. As displayed in **Fig. R9**, N117 membranes showed a low iodine permeability ($P_{N117} = 2.6014 \times 10^{-7} \text{ cm}^2 \text{ min}^{-1}$). Although the migration of negatively charged iodine species is supposed to be rejected by the Donnan exclusion effect of the N117 membrane, the cross-over would still diffuse through the micropore and swelling channel of the N117 membrane. On the other hand, as shown in **Fig. R10**, the Zn-I FBs using the N117 membrane (2 mL of 2 M ZnI_2 || 8 mL of 2 M ZnCl_2 at 30 mAh) showed inferior performance with CE, VE, and EE of 99-75-74%, 99-66-65%, 99-56-55%, 99-48-48% and 99-41-40% at 7.5, 15, 22.5, 30 and 37.5 mA cm^{-2} , respectively. This inferior performance can be attributed to the low ionic conductivity of the dense N117 membrane. In contrast, due to the high ionic conductivity of porous PP membrane and large-size starch-based active species, the flow-mode Zn-IS FBs using the PP membrane exhibited superior rate cycling and performance with high CE, VE and EE of 94-91-86, 96-85-83%, 98-81-78%, 98.5-75-74% and 98.6-70-70% at 7.5, 15, 22.5, 30 and 37.5 mA cm^{-2} , respectively (**Fig. R11, Fig. 3a & b** in the manuscript). We are sorry for not providing these results about the N117 membrane in our original manuscript. According to your suggestion, we have provided them on **Page 10** in the revised manuscript and **Supplementary Fig. 26 (Page S28), Supplementary Fig. 30 (Page S32)**.

Fig. R9 | (a) Photographs of the KI_x permeate solutions through N117 membranes under 2 M KI_x . (b) UV-vis of the KI_x permeated side. (c) $-\ln(1-C_B/C_A)$ vs. permeation time for the determination of permeability of KI_x through N117 membranes under 2 M KI_x . The fits in the $-\ln(1-C_B/C_A)$ vs. t (time) plots in (c) were obtained by linear fitting.

Fig. R10 | (a) CE, VE and EE of the Zn-I FBs cell (2 ml of 2 M ZnI₂ || N117 membranes || 8 ml of 2 M ZnCl₂, 4 cm² membrane area) under 7.5, 15, 22.5, 30 and 37.5 mA cm², (b) Voltage profiles of the Zn-IS FBs under 7.5, 15, 22.5, 30 and 37.5 mA cm².

Fig. R11 | (a) CE, VE and EE of the Zn-IS FBs cell (2 ml of 2 M ZnI₂ with 1 M starch || PP membranes || 8 ml of 2 M ZnCl₂, 4 cm² membrane area) under 7.5, 15, 22.5, 30 and 37.5 mA cm², (b) Voltage profiles of the Zn-IS FBs under 7.5, 15, 22.5, 30 and 37.5 mA cm².

Comment 8: On page 10, Fig. 4a, at a current density of 7.5 mA cm⁻², CE decreases as the number of cycles increases from 0 to 6. The authors should explain why it is not stable and decreases.

Response: Thanks for your insightful comment. I⁻ in solution was easily oxidized to I₃⁻ in the presence of air and light irradiation, according to the previous reports (*Angew.Chem.* 2018, **130**,11341; *J. Am. Chem. Soc.*, 2009, **44**, 16206; *Adv. Funct. Mater.* 2023, **33**, 2303225). As depicted in **Fig. R12**, the oxidation rate will increase at high temperatures (50 °C). During the Zn-I FBs flow cells operating at

high temperatures (50 °C) in our work, the electrolyte was pre-flowing within the battery for 1 hour to ensure the electrolyte infiltration into the reaction components of flow battery systems. This process may lead to the oxidation of iodide ions. Therefore, high CE showed in the initial cycles of the battery and then decreased as the number of cycles increased from 0 to 6, which can be attributed to the oxidized I⁻ acting as the active species to supply the discharging capacity. We are sorry for not providing this information in our original manuscript, and it was discussed in the revised manuscript (**Page 12**).

Fig. R12 | Photographs of 2 M ZnI₂ solution placed for 1 hour under 50 °C.

Comment 9: *On page 10, "The results could be attributed to the ultrasmall colloidal starch, which could cross the membrane to the anolyte and consequently stabilize the pH of the anolyte, thus conferring improved reversibility of the Zn anode." The authors should provide more detailed mechanisms or references for this statement.*

Response: We appreciate your insightful comments. Firstly, we have carefully concluded the overall problems of Zn metal anodes (**Fig. R13**) to elucidate the mechanism of colloidal starch further. As shown in **Fig. R13a**, the reactions at the Zn metal anode side are significantly influenced by the pH of electrolytes. Specifically, during the electrodeposition process of the Zn, the original hydrated Zn²⁺(H₂O)_n clusters in pure Zn²⁺ electrolyte (**Fig. R13b**) will produce a large number of active H₂O molecules at the interface between Zn metal and electrolyte (Angew. Chem. 2022, 134, e202112304; Angew. Chem. 2021, 133, 18395; J. Phys. Chem. Lett. 2018, 9, 6683), which results in the following reactions:

- a) Active H₂O molecules at the Zn anode surface can decompose into H⁺ and OH⁻;
- b) The accumulated H⁺ ions are reduced into H₂ by obtaining electrons from Zn metal and escaping from the electrolyte, leading to the corrosion of Zn metal. Meanwhile, H₂ generation also occurs during electrochemical processes;

- c) On the other side, the remaining OH^- groups after H_2 evolution can elevate the pH value inside the whole electrolyte, and the surface of Zn metal is inclined to experience passivation due to the formation of insulating by-products (mainly basic zinc salt precipitation).

Therefore, maintaining a stable pH at the interface or within the electrolyte of the zinc metal anode system could effectively mitigate issues such as corrosion and other undesirable side reactions.

Fig. R13 | (a) Potential-pH Pourbaix diagram of the Zn/water system developed using guidelines. (b) Schemes illustrating different reaction processes of Zn^{2+} solvation structure and corresponding interfacial interaction between Zn anode surface and electrolyte.

On the other hand, according to research in Zn-based coin cells (*Nat Commun*, 2022, **13**, 5348; *J. Am. Chem. Soc.* 2022, **25**, 11129), some additives in aqueous colloid electrolytes as fast ion carriers can modulate the typical electrolyte system for improving reversible plating/stripping on Zn anode for high-performance Zn ion batteries. The results were attributed to the suppressing H_2 generation of the colloidal electrolyte additives and the reduction in Zn nucleation.

In this work, as depicted in **Fig. R14**, the catholyte contains small-sized starch particles capable of migrating into the anolyte, which would form a colloidal electrolyte to a certain extent. To evaluate colloidal starch-based electrolytes for improving cyclic stability, voltage profiles of Zn plating/stripping were studied based on asymmetric FBs (Zn foil@Carbon felt || PP membranes || Carbon felt) in **Fig. R15**. The Zn-based asymmetric FBs with colloidal starch (**Fig. R15b**) exhibited a prolonged lifespan over 240 h (340 cycles) at 30 mA cm^{-2} under 10 mAh cm^{-2} , which is longer than that of the asymmetric FBs in blank electrolytes (over 150 h, **Fig. R15a**). As displayed in the inset of **Fig. R15a**, Coulombic efficiency (CE) of Zn-based asymmetric FBs in blank 2 M ZnCl_2 initially remained at 98.23% and then sharply dropped after 140 h of operation due to soft-short circuits caused by side reactions and zinc dendrites. However, the CE of Zn-based asymmetric FBs in starch-based electrolytes reached 99.01% in the inset of **Fig. R15b**, which indicated the colloidal electrolyte could effectively promote reversibility in the Zn deposition/dissolution reaction. Moreover, the variation in pH of the anolyte before and after the reaction

can further validate these results (the inset of **Fig. R15**, tested by pH meter (Brand: REX, Model: PHBJ-261L)). The pH of blank electrolytes increased from 4.42 to 4.75 after 70 h and reached 5.13 after 140 h of operation, which can be attributed to excessive side reactions in the blank electrolyte, resulting in the inferior reversibility of the Zn metal anode. In contrast, the FBs systems with the starch-based electrolyte maintained a stable pH value throughout prolonged cycles in the FB operation.

Furthermore, SEM images and XRD results under the electrolytes with/without starch after 30 cycles from Zn-I FBs were further exhibited in **Fig. R16 & Fig. R17**. The results of the membrane and anodic electrode showed a dendrite-free and clear surface after cycles based on the colloidal starch-based electrolytes. In contrast, all components based on the blank electrolytes revealed side-products of $Zn_5(OH)_8Cl_2 \cdot H_2O$. Hence, the colloidal electrolyte effectively stabilizes the reaction milieu of the anolyte, consequently enhancing the reversibility of zinc deposition/dissolution processes.

We are sorry for not providing these discussions in our original manuscript. According to your suggestion, the related discussions have been provided on **Pages 10 & 11** in the revised manuscript and **Supplementary Fig. 32 - Fig. 36 (Pages S34 - S38)**.

Fig. R14 | Size distribution of the colloidal particles in blank 1 M starch solution and 1 M starch at 50% SOC.

Fig. R15 | Long cycling of Zn-based asymmetrical FBs (Zn foil@Carbon felt || PP membranes || Carbon felt) with (a) blank 2 M ZnCl₂ and (b) 2 M ZnCl₂ with colloidal 1 M starch additives at 10 mAh cm⁻² under 30 mA cm⁻². The insets show the representative CE, charging-discharging polarization curves, and pH.

Fig. R16 | SEM images of (a) pristine CF, (b) CF anode using PP membrane without starch, and (c) CF anode using PP membrane with starch in the discharging state at 22.5 mA cm^{-2} and 33.5 Ah L^{-1} after 30 cycles. (d) XRD pattern of pristine CF, CF using PP membrane without starch, and CF using PP membrane with starch on the anodic side in the discharging state at 22.5 mA cm^{-2} and 33.5 Ah L^{-1} after 30 cycles, wherein PDF card of $\text{Zn}_5(\text{OH})_8\text{Cl}_2 \cdot \text{H}_2\text{O}$ and ZnO by-products are #07-0155 and #36-1451.

Fig. R17 | SEM images of (a) pristine PP membrane, (b) PP membrane without starch and (c) with starch in the discharging state at 22.5 mA cm^{-2} and 33.5 Ah L^{-1} after 30 cycles. (d) XRD pattern of pristine PP

membrane, PP membrane with/without starch after cycles in the discharging state at 22.5 mA cm^{-2} and 33.5 Ah L^{-1} after 30 cycles, wherein PDF card of $\text{Zn}_5(\text{OH})_8\text{Cl}_2 \cdot \text{H}_2\text{O}$ is #07-015.

Comment 10: *The authors should provide permeation test data for 50 °C conditions.*

Response: We appreciate your helpful and valuable suggestions. According to your suggestions, we further evaluate the permeability of IS colloids across the PP membrane. Two-compartment H-cells consisting of the IS catholyte at a 50% state of charge in one cell and deionized water in another cell were used under high temperature (50 °C), as schemed in **Fig. R18**. The UV-visible spectra and permeation rate of nominally prepared I_x^- solutions under different starch concentrations (**Supplementary Fig. 4** in supporting information), wherein the strong peaks with the absorption wavelength of 288 and 350 nm are attributed to the presence of I_3^- . Specifically, the I_x^- permeability largely decreased based on starch electrolytes compared to the severe permeability in blank electrolytes, as shown in **Fig. R19 & Fig. R20**. It indicated that the colloidal starch could strongly confine the polyiodides by forming a colloidal aggregation featuring low I_x^- permeability to impede the cross-over issue even at a severe high-temperature condition.

We are sorry for not providing these results in our original manuscript, and the related discussions have been provided on **Page 12** in the revised manuscript and **Supplementary Fig. 41 & 42 (Pages S43 & S44)**.

Fig. R18 | Photographs of two-compartment H-cell testing configuration in an incubator at a high temperature (50 °C).

Fig. R19 | (a) Photographs of the KI_3 permeate solutions through PP membranes under blank 2 M KI_3 electrolytes at high temperature of $50 \text{ }^\circ\text{C}$. (b) UV-vis of the KI_3 permeated side at high temperature of $50 \text{ }^\circ\text{C}$. (c) $-\ln(1-C_B/C_A)$ vs. permeation time for the determination of permeability of KI_3 through PP membranes under blank 2 M KI_3 at high temperature of $50 \text{ }^\circ\text{C}$. The fits in the $-\ln(1-C_B/C_A)$ vs. t (time) plots in (c) were obtained by linear fitting.

Fig. R20 | (a) Photographs of the KI_x permeate solutions through PP membranes under 2 M KI_x with 1 M starch electrolytes at high temperature of $50 \text{ }^\circ\text{C}$. (b) UV-vis of the KI_x permeated side at high temperature of $50 \text{ }^\circ\text{C}$. (c) $-\ln(1-C_B/C_A)$ vs. permeation time for the determination of permeability of KI_x through PP membranes under 2 M KI_x with 1 M starch at high temperature of $50 \text{ }^\circ\text{C}$. The fits in the $-\ln(1-C_B/C_A)$ vs. t (time) plots in (c) were obtained by linear fitting.

Comment 11: On page 10, Fig. 4e, at cycle number from 0 to 50, the authors should explain why the volume capacity increases as the cycle number increases. It looks different from the electrochemical performance at 25 °C.

Response: Thanks for your insightful and valuable comments. Both side reactions of anodes and cross-over of active materials would be aggravated at high temperatures (*Nano Lett.*, 2022, 4, 1549; *ACS Energy Lett.*, 2023, 3, 1613), i.e., 50 °C in our work, while the reaction kinetics in cathodes would be enhanced (*Nat Commun*, 2020, 11, 3204.) due to an activation process of the iodine side. On the other hand, there were more side reactions occurring during the Zn deposition/dissolution process of the initial stage, resulting in HER and leaving behind the incomplete dissolved Zn, which can be demonstrated by SEM image after cycles at 50 °C condition (**Fig. R21**). However, those undissolved Zn accumulate and retains the electrochemical activity (*Energy Environ. Sci.*, 2023, 16, 438; *ACS Appl. Mater. Interfaces*, 2022, 14, 51010; *J. Am. Chem. Soc.* 2023, 145, 24284.). Consequently, the undissolved Zn can be further dissolved and contribute supplementary Zn metal for subsequent cycles to improve the CE.

Therefore, the increasing capacity of the as-developed Zn-I FBs in the initial stage can be attributed to the activation of the FBs, and then the Zn-I FBs systems would gradually change to the reaction steady state. We are sorry for not providing this information in our original manuscript, and it was discussed in the revised manuscript on **Page 12** and **Supplementary Fig. 45 (Pages S47)**.

Fig. R21 | SEM images of CF anode using PP membrane with starch in the discharging state after 30 cycles at 50 °C condition.

Responses to Reviewer's Comments

Reviewer #2:

The authors utilized starch to form complexes with iodine ions, effectively increasing the molecular size. This approach was aimed at preventing cross-over and enabling the use of more cost-effective hydrocarbon membranes in the Zn-I flow battery system. Their results demonstrated stable cycling performance when starch was used as an additive. However, there are several areas where the manuscript could be improved.

In conclusion, this study is intriguing, but it would greatly benefit from revisions in the mentioned areas. I encourage the authors to make these improvements and resubmit their manuscript for further consideration.

General Response: Thanks for your professional assessment of our work. We appreciate your positive evaluations of the enhancements for cost-effective Zn-I flow batteries in cycling performance in our work concerning starch-based electrolytes. We have carefully studied your comments and addressed them as follows.

Comment 1: *While the authors explored the impact of starch on the physical properties of the iodine electrolyte, they did not conduct an electrochemical study to investigate how starch affects the crucial thermodynamic redox behavior and kinetics of the iodine/iodide couple. This is an essential aspect that should be addressed.*

Response: Thanks for your constructive and valuable comments. Following your comments, we used some electrochemical measurement methods (chronoamperometry (CA) and cyclic voltammetry (CV)) to evaluate the colloidal starch's effect on the electrochemical reaction of the iodine/iodide couple redox. Here we discussed it as follows.

a) To explore the effect of starch on the redox reaction of iodine, we conducted a chronoamperometry (CA) test using the KI electrolytes with/without starch at a constant potential of 1.2

V (**Fig. R1**). For the blank KI solutions, it is observed that the current experiences a sharp decrease in the first two seconds, followed by a steady-state value of approximately 1.01 mA. The decline can be ascribed to the formation of solid I_2 that covers the electrode, passivating the reaction of the active material at the initial reaction process (*J. Mater. Chem. A*, 2022, **10**, 14090). For the KI electrolytes with starch, the current showed a much larger value of 7.29 mA and remained constant during the CA test, which can be demonstrated that starch could rapidly interact with products (I_2/I_3^-), continuously uncovering the fresh surface of the electrode, which favors the redox reactions both thermodynamically and kinetically.

Fig. R1 | Chronoamperograms in 0.1 M KI and 0.1 M KI with 1 M starch.

b) To simulate a real flow battery environment, stirring was applied within the reactor to ensure the thorough dispersion of the oxidized I_3^- into the electrolyte (**Fig. R2a**). The cyclic voltammetry (CV) curve of I_3^-/I^- redox reactions in potassium iodide (KI) electrolytes with/without starch in the potential range of -0.4 to 1.2 V (vs. SCE) was tested by the three-electrode mode. Due to the excess of iodide ions, the oxidation reaction continued, which could not show the distinct oxidation peak (Fig. R2 (b & c)). Moreover, the Tafel plots were calculated by corresponding CV curves. As shown in **Fig. R2d**, the starch-electrode was characterized by lower Tafel slopes for both oxidation (160 mV dec^{-1}) and reduction (13 mV dec^{-1}) steps than those of blank electrolytes-based electrode (oxidation: 238 mV dec^{-1} , reduction: 160 mV dec^{-1}), proving the significantly enhanced reaction kinetics of iodide species in both charging and discharging processes in Zn-I FBs. Those results can be attributed to the blocked electrode by passivating solid I_2 in blank electrolytes while keeping the clean surface of the electrode with starch.

Fig. R2 | (a) The photograph of the reactor under flowing-mode conditions. Cyclic voltammetry of redox reactions in (b) 0.1 M KI and (c) 0.1 M KI with 1 M starch under flowing-mode conditions. (d) the corresponding Tafel plots in different electrolytes.

In summary, the colloidal starch in electrolytes can facilitate the electrochemical reaction of iodide/polyiodides. Although it influences the mass transfer of active species due to the ionic conductivity, the colloidal starch was positive to sustain high electrochemical reactivity of the positive electrode in both thermodynamics and kinetics. Again, we are sorry for not providing these results in our original manuscript. Following your advice, the discussion has been provided on **Page 9** in the revised manuscript and **Supplementary Fig. 24 & Fig. 25 (Pages S26 & S27)**.

Comment 2: *The Zn-I system is known for its high energy density due to the high solubility of ZnI₂ salt. The authors only tested up to a 2M concentration of ZnI₂. It would be valuable to explore what happens when the concentration is increased to its solubility limit and whether starch remains effective. A more comprehensive study in this regard is recommended.*

Response: We appreciate your valuable and insightful suggestions. As shown in **Fig. R3**, an increased concentration of starch can anchor more polyiodides while leading to higher viscosity and decreased ionic conductivity. For high-concentration ZnI₂ salts, utilizing higher concentrations of polyiodides requires higher concentrations of starch for fixing under the high-concentration ZnI₂ salts, which may lead to inferior power density for Zn-I FBs. In contrast, when high concentrations of iodine are fixed by low concentrations of starch, the cross-over of polyiodides will be exacerbated to result in poor CE, explained in detail below.

According to your suggestion, given that increasing the concentration of ZnI₂ salt can enhance the energy density of flow batteries, we attempted to evaluate the electrochemical performance of Zn-I FBs operating under a 6 M ZnI₂ solution at 40% SOC with/without 1 M starch. As displayed in **Fig. R4**. The Zn-I FBs with blank electrolytes at 30 mA cm⁻² exhibited inferior Coulombic efficiency (CE) at around

83% (77.1 Ah L^{-1}) and gradually increasing polarizations on account of severe cross-over and anodic side reactions (**Fig. R4b**). In contrast, The Zn-I FBs with colloidal starch-based electrolytes delivered a stable charge-discharge operation at the same conditions with stable CE (average $>92\%$), realizing the volumetric capacity of $84.1 \text{ Ah L}^{-1}_{\text{catholyte}}$, which could demonstrate the effective inhibition on the cross-over issue of iodine species (**Fig. R4 (a & c)**). Therefore, compared to the pristine Zn-I FBs without starch, colloidal starch could inhibit the shuttling of polyiodides under high concentrations to deliver higher CE and volumetric capacity due to the enhanced size-sieving effect. However, using 6 M ZnI_2 concentration resulted in a lower CE (92%) compared to 2 M (CE of 98%), leading to a reduced Energy efficiency (EE). Additionally, owing to the hybrid Zn-I FBs, the Zn anode faced a more severe dendritic zinc challenge at the 6 M concentration, resulting in a shorter cycle life than that of the 2 M concentration.

After carefully considering critical performance metrics encompassing energy efficiency and cycle life, we selected 1 M starch and 2 M ZnI_2 as research objects in this work. It is worth noting that the energy density of 2 M ZnI_2 catholyte at 50% SOC is also competitive compared with other systems. Following your suggestion, we have further illustrated the consideration of the starch concentration on electrochemical performance in the revised manuscript and the **NOTE of Supplementary Fig. S23 (Page S25)**.

Fig. R3 | Viscosity, ionic conductivity, and I_x^- permeability of 2 M ZnI_2 electrolytes with different starch concentrations ($0, 0.1, 0.2, 0.5, 1, 2,$ and 3 M).

Fig. R4 | (a) Cycling performances of Zn-IS FBs flow-cell system with/without starch (2 ml of 6 M ZnI₂ with 1 M starch || PP membranes (two layers) || 8 ml of 6 M ZnCl₂, 4 cm² membrane area) at high volume capacity (50% SOC, 90 Ah L⁻¹) under a current density of 22.5 mA cm⁻². The corresponding voltage profiles of Zn-IS FBs flow-cell system (b) without or (c) with starch.

Comment 3: Figure 3d shows a significant difference in the capacity of the cell with and without starch. This difference needs a clear explanation. It should be clarified whether these tests were conducted at the same concentration, and if not, the reasons for the difference should be provided.

Response: Thanks for your insightful suggestions. We sincerely apologize for not providing a sufficient description of the performance differences in our manuscript, and we have reorganized the content in the following sections to ensure better comprehension by the reviewers.

Firstly, the electrochemical performances of Zn-I FBs using PP membrane with/without starch were tested at the same condition in our manuscripts. Specifically, as shown in **Fig. R5d** (**Fig. 3d** in the manuscripts), Zn-IS FBs delivered a stable charge-discharge operation over 350 cycles at a high current density of 30 mA cm⁻² with high CE (98.5%), realizing the volumetric capacity of 6 Ah L⁻¹_{catholyte}. To emphasize the role of starch in suppressing the issues of polyiodide cross-over, Zn-I FBs using PP membrane with/without starch based on high-energy-density conditions were conducted for comparison (**Fig. 3e** in the manuscripts). Under a high volumetric capacity (33.5 Ah L⁻¹_{catholyte}) to achieve 50%

utilization of the iodide, i.e., 50% SOC, this Zn-IS FBs flow system demonstrated long cycling life (over 250 cycles) with a high CE (~95%) at 22.5 mA cm⁻² in **Fig. R5e (Fig. 3e** in the manuscripts), indicating the effective inhibition on the cross-over issue of iodine species. Under the same working condition, the PP membrane-based flow batteries in blank electrolytes without starch showed inferior CE at around 65% with severe capacity loss, lower discharging capacity as ~25 Ah L⁻¹_{catholyte}, and short cycle lifespan (~50 cycles) due to the severe cross-over and short-circuits (**Fig. R6, Supplementary Fig. 29** in the manuscript). Furthermore, we conducted further analysis of the reason for the performance difference between the Zn-I FBs with and without starch, which can be attributed to two points as following details:

Based on the high SOC (50%) operating condition, the prolonged discharge duration of the Zn-I FBs will exacerbate the cross-over of polyiodides in non-starch systems. Subsequently, the active Zn metal in the anode will be consumed by that I₃⁻, further leading to a decrease in CE and exacerbating low reversibility issues of Zn deposition/dissolution. Secondly, under a high areal capacity in Zn-based FBs, the anode will exacerbate side reactions and dendrite formation, resulting in inferior cycling performance of the Zn-I FBs, which can be due to the absence of zincophilic sites and stable environments for the process of Zn metal deposition/dissolution. These results can also be demonstrated by the SEM images and XRD under the electrolytes without starch after cycles from Zn-I FBs, as shown in **Fig. R7**.

On the other hand, according to research in Zn-based coin cells (*Nat Commun*, 2022, **13**, 5348; *J. Am. Chem. Soc.* 2022, **25**, 11129), some additives in aqueous colloid electrolytes as fast ion carriers can modulate the typical electrolyte system for improving reversible plating/stripping on Zn anode for high-performance Zn ion batteries. The results could be attributed to the colloidal electrolyte additives (not starch) in reducing nucleation overpotential and activation energy of Zn plating and suppressing H₂ generation. As depicted in **Fig. R8**, small-sized starch particles in the catholyte can migrate into the anolyte, forming a colloidal electrolyte on the anolyte to a certain extent. SEM images and XRD results of the membrane and anodic electrode showed a dendrite-free and clear surface after cycles based on the colloidal starch-based electrolytes from Zn-I FBs (**Fig. R9**). Hence, the colloidal electrolyte effectively stabilizes the reaction of the anolyte, consequently enhancing the reversibility of zinc deposition/dissolution processes.

In summary, the substantial differences in performance in our manuscript can be attributed to the cross-over of active polyiodine and the reaction reversibility of the Zn metal. Following your suggestion, we have further discussed these two points on **Page 10 & 11** in the revised manuscript and the corresponding **Supplementary Fig. 32 - Fig. 36 (Page S34 - 38)**.

Fig. R5 | Electrochemical performance of the Zn-IS FBs at 25 °C. (a) CE, VE and EE of the Zn-IS FBs cell (2 ml of 2 M ZnI₂ with 1 M starch || PP membrane || 8 ml of 2 M ZnCl₂, 4 cm² membrane area) under 7.5, 15, 22.5, 30 and 37.5 mA cm⁻², (b) Voltage profiles of the Zn-IS FBs under 7.5, 15, 22.5, 30 and 37.5 mA cm⁻². (c) The polarization of the Zn-IS FBs using different membranes (PP with 1 M starch & N117 without starch). (d) Cycling performances of Zn-I FBs flow-cell system at high power density (30 mA cm⁻²). (e) Cycling performances of Zn-IS FBs flow-cell system with/without starch at high volume capacity (50% SOC, 36 Ah L⁻¹) under a current density of 22.5 mA cm⁻². The inset in (d) & (e) are the corresponding voltage profiles.

Fig. R6 | (a) Cycling performances of Zn-I FBs flow-cell system using PP membrane without starch at high volume capacity (33.5 Ah L^{-1}) under 22.5 mA cm^{-2} . Selected cycles corresponding to (b) region (18 th -23 th) and (c) region (54 th - 56 th) in (c), where the short-circuit point and cross-over are marked.

Fig. R7 | SEM images of (a) pristine CF, (b) CF anode using PP membrane without starch, and (c) CF anode using PP membrane with starch in the discharging state at 22.5 mA cm^{-2} and 33.5 Ah L^{-1} after 30 cycles. (d) XRD pattern of pristine CF, CF using PP membrane without starch, and CF using PP

membrane with starch on the anodic side in the discharging state at 22.5 mA cm^{-2} and 33.5 Ah L^{-1} after 30 cycles, wherein PDF card of $\text{Zn}_5(\text{OH})_8\text{Cl}_2 \cdot \text{H}_2\text{O}$ and ZnO by-products are #07-0155 and #36-1451.

Fig. R8 | Size distribution of the colloidal particles in blank starch solution and starch at 50% SOC

Fig. R9 | SEM images of (a) pristine PP membrane, (b) PP membrane without starch and (c) with starch in the discharging state at 22.5 mA cm^{-2} and 33.5 Ah L^{-1} after 30 cycles. (d) XRD pattern of pristine PP membrane, PP membrane with/without starch after cycles in the discharging state at 22.5 mA cm^{-2} and 33.5 Ah L^{-1} after 30 cycles, wherein PDF card of $\text{Zn}_5(\text{OH})_8\text{Cl}_2 \cdot \text{H}_2\text{O}$ is #07-015.

Comment 4: *The inclusion of a cost analysis is commendable, but the study would benefit from a more comprehensive assessment of the overall system cost, including the cost of chemicals. This additional information would provide readers with a more precise understanding and facilitate comparisons with other systems.*

Response: We appreciate your constructive suggestions. According to your suggestion, we have incorporated the chemical cost into the overall system cost. To improve a more precise understanding and facilitate comparisons with other systems, the cost of chemicals was analyzed in **Supplementary Table 2** for a 1 MW zinc-iodine flow battery stack with 1 MWh. Meanwhile, considering the practical Zn-I FBs stack, the detailed cost analysis was conducted at the power realized at long-term operating current density (22.5 mA cm^{-2}) based on this performance obtained in our work.

Specifically, regarding the power realized at 22.5 mA cm^{-2} and the long-term cycling performance in Fig. 3e (manuscript), the stack area of N117 membrane-based Zn-I FBs needs about 5012 m^2 at 19.95 mW cm^{-2} for working 1 MW Zn-I FBs. The stack area of PP membrane-based Zn-IS FBs only needs about 3766 m^2 at 26.55 mW cm^{-2} for working 1 MW Zn-IS FBs. The detailed cost and the amount of components for the 1-MW flow cell stack are provided in **Supplementary Table 1**. Specifically, the Nafion membrane cost accounted for the highest proportion (88.49%, $\$500 \text{ per m}^2$), leading to the high installation cost of the full cell as 1.97 million dollars for the 1-MW cell. By contrast, by replacing the Nafion 117 membrane with the PP membrane, the cost of the 1-MW flow stack dramatically decreased by 90.02% in the installed cost to 0.28 million dollars, *i.e.*, 14.3 times lower in cost. As shown in **Supplementary Table 2**, we further analyzed the cost of 1 MW system Zn-I FBs for a long-duration output of 1 MWh. Benefiting from the excellent output power in PP membranes-based Zn-IS FBs systems, the simulated Zn-IS stack (total 0.636013 \$ million) presented a lower price in a 1 MWh energy storage system than N117-based Zn-I FBs stack (total 3.184189 \$ million).

Following your suggestion, we have included the cost of chemicals and provided the detailed cost analysis on **Page 14** in the revised manuscript and the corresponding **Supplementary Table 1 & 2 (Page S51 & S52)**.

Supplementary Table 1 | The components and their costs for a 1 MW zinc-iodine flow battery stack

Component	Price per unit (\$ m²)	Amount of N117 membrane-based Zn-I FBs (m²)	Total cost of N117 membrane-based Zn-I FBs (\$)	Amount of PP membrane-based Zn-IS FBs (m²)	Total cost of PP membrane-based Zn-IS FBs (\$)
------------------	--	---	--	--	---

Bipolar plates	30	5012	150360	3766	112980	
Membrane	PP: 10		2506000		3766	37660
	N117: 500					
Frame	15		75180		56490	
Carbon felt	20		100240		75320	
Gasket-1	1	4 pieces	4	4 pieces	4	
Gasket-2	2	4 pieces	8	4 pieces	8	
Total cost for 1 MW Zn-I FBs		2.831792 million (\$)		0.282462 million (\$)		

Supplementary Table 2 | The components and their costs for a 1 MW zinc-iodine flow battery stack with 1 MWh.

Chemicals	Molecular weight (g mol⁻¹)	Price per unit (\$ kg⁻¹)	Amount of chemicals (kg)	The cost of chemicals (\$)
ZnCl₂	136.28	11	6934.1	0.076275 million
ZnI₂	319.22	17	16242.4	0.276122 million
Starch	324.28	0.14	8249.9	0.001154 million
Total cost for 1 MW N117 membrane-based Zn-I FBs with 1 MWh (stack + chemicals)			3.184189 million (\$)	
Total cost for 1 MW PP membrane-based Zn-IS FBs with 1 MWh (stack + chemicals)			0.636013 million (\$)	

Responses to Reviewer's Comments

Reviewer #3:

Development of low-cost and efficient energy storage technologies are of key importance for realizing the energy transition towards a clean model based on renewable sources. Among the various emerging technologies, aqueous zinc-iodine flow batteries (Zn-I FBs) holds great promising due to its low cost and the independent scalability of power and energy. However, there are several issues that prevent practical implementation of this promising technology. In this context, the authors report a strategy to avoid the use of expensive ion-selective membrane while improving the energy efficiency. They propose the replacement of ion-selective membrane by porous separators. While this strategy has been long desired, porous separator leads to cross-contamination of positive and negative species, which results in short lifespan. The proposed strategy is based on the addition of starch which results in a complexation with polyiodide species. This complexation increases the size of the active species, so that size exclusion porous separators are able to prevent the cross-contamination. The hypothesis is sound, and the findings reported in the manuscript seem to support the hypothesis. The demonstrated power density and specially the lifespan are very relevant in this field. Overall, this work could become suitable for publication in Nature Communication as long as the authors are able to address four major issues:

General Response: Thank you for the insightful evaluation of our work. We appreciate your insightful assessments regarding the improvements made for cost-effective Zn-IS flow batteries in the performance of power density and specially the lifespan using starch-based electrolytes. We have carefully studied your comments and revised our manuscript according to address them.

Comment 1: *The internal resistance when ion-selective membrane is used is claimed to be drastically increase with respect to the proposed strategy. As a result, the power density achieved using N117 Nafion membrane was only of 28 mW cm^{-2} against 70 mW cm^{-2} for 1 M starch. The EIS measurements are used to explain the poor power density. However, the difference for Nafion and PP separators is only 0.16*

Ohm cm⁻². At 40 mA cm⁻², the voltage drops induced by such a difference would be 4.5 mV. Thus, charge transfer resistance and diffusion must be the origin. However, the low frequencies resistance including all parameters is even lower than 0.16 Ohm cm⁻², which is due to the presence of starch. Therefore, the differences in power density are not supported by the results. This point must be explained properly.

Response: Thanks for your constructive suggestions. We agree with your assertion that using the EIS measurements to substantiate the improved power density might not be rational. Furthermore, we sincerely apologize for the mistake in the EIS results, where the correct unit for EIS should be Ohm cm² rather than Ohm cm⁻² (*Nat. Mater.*, 2020, **19**, 195; *Nat Commun*, 2022, **13**, 3184), as shown in **Fig. R1**. Additionally, it should be clarified that the power density of Zn-I FBs obtained using starch-based electrolyte with a PP membrane at room temperature is 41.58 mW cm⁻², rather than 70 mW cm⁻². Nevertheless, the EIS results can present the impedance measured by applying alternative currents (AC) within a wide frequency range, while the EIS results could not represent the internal resistance of a flow battery under realistic working conditions by applying direct currents (DC).

Different DC current densities were applied on the Zn-I FBs at a 50% SOC to measure the internal resistance of the flow batteries. As shown in **Fig. R2**, the polarization voltage obtained from charge-discharge based on a short working time (1 min) at different current densities of 10-40 mA cm⁻² were fitted to calculate the internal resistance values. The internal resistance of Zn-IS FBs with starch using a PP membrane is 4.08 Ω cm², which is remarkably lower compared to Zn-IS FBs without starch using an N117 membrane (10.94 Ω cm²). The internal resistance difference at the cell level was mainly responsible for the power difference of these two types of Zn-IS FBs, while the remaining difference could be assigned to the resistance variations caused by activation polarization and concentration polarizations.

We apologize for not discussing this point in our original manuscript. Following your comment, we have cautiously revised the discussions about the EIS section and added the relevant internal resistance under realistic working conditions for these two types of Zn-IS FBs. Please see the highlighted section on **Page 9** in the revised manuscripts and the corresponding **Supplementary Fig. 27 & Fig. 28 (Pages S29 & S30)**.

Fig. R1 | EIS of the Zn-I FBs flow cell using different membranes (PP membranes with 1 M starch & N117 membranes without starch) under charging to 50% SOC.

Fig. R2 | The polarization voltage of the Zn-I FBs flow cell using different membranes ((a) PP membranes with 1 M starch & (b) N117 membranes without starch) under charging to 50% SOC at different current density (10 - 40 mA cm^{-2}).

Comment 2: *Cross over of water due to osmotic issues. Having asymmetric electrolyte will have an impact in the viscosity and osmotic properties. The transfer of water between the two compartments is likely to occur in the long run. This point needs to be discussed as it may hinder implementation. How do the authors propose to overcome this relevant issue?*

Response: Thanks for your valuable and insightful suggestions. The asymmetric electrolyte-driven water migration remains a long-addressed challenge for flow batteries. This problem is particularly severe in dense ion exchange membranes, where the water migration occurs via hydrated ions during the ion exchange process (*Nat Energy*, 2021, **6**, 517; *Nature Mater*, 2008, **7**, 75; *J. Membr. Sci.*, 2003, **222**, 235). As shown in **Fig. R3a**, the supplementary experiment involving polyiodide permeation in the H-type cell based on dense N117 membranes demonstrated a severe water out-of-balance issue within the H-type cell after 24 h testing.

On the other hand, benefiting from the porous hydrophilic PP membrane utilized in this work, the hydrated ions and the water molecules can accommodate free movement (*Nat Energy*, 2016, **1**, 1609; *Nano Lett.*, 2021, **21**, 10446). As depicted in **Fig. R3(b & c)**, the H-type cell based on the PP membrane showed no significant water migration on both sides after a 24-hour permeation experiment with/without starch additives. This result can be attributed to the free movement of the ions and the H₂O molecules across the porous hydrophilic PP membrane (the pore diameter of 37.28 nm), which could balance and stabilize the osmotic pressure of the electrolyte.

For this work, the initial stage of water migrated into the starch-based cathode electrolyte due to the lower volume of the catholyte and the high concentration of starch. Subsequently, as mentioned in features of porous PP membranes, the volume of catholytes and anolytes tended to balance as FBs cycles. We apologize for not discussing this point in our original manuscript. According to your comment, we have further discussed the advantage of applying a PP membrane to inhibit the water cross-over issue. Please see the highlighted section in the corresponding revised **Supplementary Fig. 23 (Page S25)**.

Fig. R3 | (a) Photographs of the KI_x permeate solutions through N117 membranes under 2 M KI_x after 24 h. (b) Photographs of the KI_x permeate solutions through porous PP membranes under 2 M KI_x after 24 h. (c) Photographs of the KI_x permeate solutions through porous PP membranes under 2 M KI_x with 1 M starch electrolytes after 24 h.

Comment 3: *Coulombic efficiency. The value of Coulombic efficiency is 98 %. Considering the low permeability value, and the relatively short length of one cycle, cross over of polyiodide does not seem to be the source of the Coulombic inefficiency. It is necessary to estimate the impact of the cross-contamination in one cycle. If the inefficiency is coming from the negative side, how can the systems operate for having such a large amount of HER and pH change? This point requires proper explanations.*

Response: We appreciate your valuable and insightful comments. Following your suggestion, we further investigated the cross-over of active materials and the reaction reversibility of the Zn metal anode, confirming that the inefficiency was jointly attributed to the shuttling of active materials (I_x^-) and the side reactions at the Zn anode.

Specifically, on the positive side, static permeation tests were only affected by the difference in osmotic pressure and ionic strength, exhibiting lower permeation rates in **Fig. R4a** & **Fig. R5**. However, during actual flow battery testing, the driving force of an external electric field during battery operation need to be considered. There is a complete circuit during the discharge process, and the active polyiodide

anions at the positive side were driven to diffuse towards the negative side under the battery's internal electric field, exacerbating the cross-over issue of polyiodides. This result can be demonstrated through diffusion testing in an H-typed cell under an applied electric field (**Fig. R4b**). The noticeable diffusion of polyiodides from the feed side to the permeate side was observed under the influence of the applied electric field after 1 h, as displayed in **Fig. R4b**. Meanwhile, distinct characteristic peaks of polyiodides were identified by UV-vis spectrum (**Fig. R5**). In other words, the notorious cross-over issue in Zn-I FBs is influenced not only by osmotic pressure and ionic strength but also significantly by the external electric field under realistic working conditions. Therefore, the shuttling polyiodide active material and the spontaneous reaction with the Zn metal will consume Zn, resulting in Coulombic inefficiency in the Zn-I FBs systems.

Fig. R4 | Photographs of the KI_x permeate solutions through PP membranes under 2 M KI_x with 1 M starch electrolytes after 1 h under (a) static conditions and (b) an external electric field (-0.5 V), wherein two graphite rod electrodes were utilized as the positive and negative electrodes, respectively.

Fig. R5 | UV-vis of the KI_x permeated side after 1 h under (a) static conditions and (b) an external electric field (0.5 V).

Regarding the reaction reversibility at the Zn anode side, benefiting from stable colloid additives, aqueous colloid can modulate the plating/stripping behaviors of Zn anodes with improved reversibility for high-performance Zn ion batteries (*Nat Commun*, 2022, **13**, 5348; *J. Am. Chem. Soc.* 2022, **25**, 11129). As evidenced by **Fig R6**, the starch particles in the catholyte can migrate into the negative side due to their smaller size than the pore size of the PP membrane, which would form a colloidal anolyte. Furthermore, to evaluate colloidal starch-based electrolytes for improving cyclic stability, the voltage profiles of Zn plating/stripping were studied based on asymmetric FBs (Zn foil@Carbon felt || PP membranes || Carbon felt) in **Fig R7**. The Zn-based asymmetric FBs with colloidal starch (**Fig. R7b**) exhibited a prolonged lifespan over 300 h at 30 mA cm⁻² under 10 mAh cm⁻², which is longer than that of the asymmetric FBs in blank electrolytes (over 150 h, **Fig. R7a**). As displayed in the inset of **Fig. R7a**, Coulombic efficiency (CE) of Zn-based asymmetric FBs in blank 2 M ZnCl₂ initially remained at 98.23% and then sharply dropped after 140 h of operation due to soft-short circuits caused by side reactions and zinc dendrites. However, the CE of Zn-based asymmetric FBs in starch-based electrolytes was high at 99.01% in the inset of **Fig. R7b**, which indicated the colloidal electrolyte could effectively promote reversibility in the Zn deposition/dissolution reaction. Moreover, the variation in pH of the anolyte before and after the reaction can further validate these results (the inset of **Fig. R7**, tested by pH meter (Brand: REX, Model: PHBJ-261L)). The pH of blank electrolytes increased from 4.42 to 4.75 after 70 h and reached 5.13 after 140 h of operation, which can be attributed to excessive side reactions in the blank electrolyte, resulting in the inferior reversibility of the Zn metal anode. In contrast, the FBs systems with the starch-based electrolyte maintained a stable pH value throughout prolonged cycles in the FBs operation. Notably, the less than 1% deficiency in CE is not entirely attributed to HER and side reactions.

There is unreacted Zn anchored on the 3-dimensional carbon felt, which could also partially contribute to the deficiency in CE. However, we are so sorry that we cannot quantitatively analyze the amount of HER and the undissolved Zn.

Furthermore, SEM images and XRD results under the electrolytes with/without starch after 30 cycles from Zn-I FBs were further exhibited in **Fig R6 & Fig R7**. The results of the membrane and anodic electrode showed a dendrite-free and clear surface after cycles based on the colloidal starch-based electrolytes, while all components based on the blank electrolytes revealed side-products of $\text{Zn}_5(\text{OH})_8\text{Cl}_2 \cdot \text{H}_2\text{O}$. Hence, the colloidal electrolyte effectively stabilizes the reaction of the anolyte, consequently enhancing the reversibility of Zn deposition/dissolution processes.

Therefore, the Coulombic inefficiency (~98%) is attributed to two factors: one is the electromigration of the polyiodide active materials, and the other is the side reaction of the negative electrode (HER and unreacted Zn). In general, compared to Zn-I FBs with blank electrolytes, the as-developed Zn-IS FBs with colloidal starch electrolytes exhibited high power density, enhanced Coulombic efficiency, and prolonged cycling due to the improvement addressed by colloidal starch concerning the positive and negative electrode issues. We are sorry for not discussing this point in our original manuscript. Following your suggestion, these discussions have been provided on **Page 10 - 11** in the revised manuscript and the corresponding **Supplementary Fig. 32 - Fig. 36 (Page S34 - 38)**.

Fig. R6 | Size distribution of the colloidal particles in blank 1 M starch solution and 1 M starch at 50% SOC.

Fig. R7 | Long cycling of Zn-based asymmetrical FBs (Zn foil@Carbon felt || PP membrane || Carbon felt) with (a) blank 2 M ZnCl₂ and (b) 2 M ZnCl₂ with colloidal starch additives at 10 mAh cm⁻² under 30 mA cm⁻². The insets show the representative CE and charging-discharging polarization curves.

Fig. R8 | SEM images of (a) pristine CF, (b) CF anode using PP membrane without starch, and (c) CF anode using PP membrane with starch in the discharging state at 22.5 mA cm^{-2} and 33.5 Ah L^{-1} after 30 cycles. (d) XRD pattern of pristine CF, CF using PP membrane without starch, and CF using PP membrane with starch on the anodic side in the discharging state at 22.5 mA cm^{-2} and 33.5 Ah L^{-1} after 30 cycles, wherein PDF card of $\text{Zn}_5(\text{OH})_8\text{Cl}_2 \cdot \text{H}_2\text{O}$ and ZnO by-products are #07-0155 and #36-1451.

Fig. R9 | SEM images of (a) pristine PP membrane, (b) PP membrane without starch and (c) with starch in the discharging state at 22.5 mA cm^{-2} and 33.5 Ah L^{-1} after 30 cycles. (d) XRD pattern of pristine PP membrane, PP membrane with/without starch after cycles in the discharging state at 22.5 mA cm^{-2} and 33.5 Ah L^{-1} after 30 cycles, wherein PDF card of $\text{Zn}_5(\text{OH})_8\text{Cl}_2 \cdot \text{H}_2\text{O}$ is #07-015.

Comment 4: *Cost analysis. The power value used for cost analysis are peak power values, which were not shown to be feasible under continuous operation. The comparison here was conducted at 22.5 mA cm^{-2} (36 Ah L^{-1}), so that the cost analysis should be conducted at the power realized at this current density. While fair price is used for Nafion membrane, the cost of bipolar plates and graphite felt is underestimated since both are $> 50 \text{ USD m}^{-2}$ in most of the reports. As a result, the cost of the proposed systems at the demonstrated conditions will be significantly higher.*

Response: Thanks for your insightful suggestions. Considering your comments regarding practical Zn-I FBs stack, the cost analysis was corrected and conducted at the power realized at long-term operating current density (22.5 mA cm^{-2}) based on this work. Meanwhile, the cost of components of flow batteries (i.e., bipolar plates and graphite felt) fluctuated by market conditions and was analyzed again by a combination of current marketing, previous references, and recent reports (*Nat Energy*, 2021, 6, 517;

Joule, 2022, 6, 884; <https://www.alibaba.com/>). Moreover, to improve a more precise understanding and facilitate comparisons with other systems, the cost of chemicals was analyzed in **Supplementary Table 2** for a 1 MW zinc-iodine flow battery stack with 1 MWh.

According to your valuable suggestion, the cost analysis was conducted at the power realized at 22.5 mA cm^{-2} in **Fig. 3e** of the manuscript. More specifically, the stack area of N117 membrane-based Zn-I FBs needs about 5012 m^2 at 19.95 mW cm^{-2} for working 1 MW Zn-I FBs. The stack area of PP membrane-based Zn-IS FBs need about 3766 m^2 at 26.55 mW cm^{-2} for working 1 MW Zn-IS FBs. The detailed cost and the amount of components for the 1 MW flow cell stack are provided in **Supplementary Table 1**. The Nafion membrane cost accounted for the highest proportion (88.49%, \$500 per m^2), leading to the high installation cost of the full cell as 1.97 million dollars for the 1-MW cell. By contrast, by replacing the Nafion 117 membrane with the PP membrane, the cost of the 1 MW flow stack dramatically decreased by 90.02% in the installed cost to 0.28 million dollars, *i.e.*, 14.3 times lower in cost. As shown in **Supplementary Table 2**, we further analyzed the cost of 1MW system Zn-I FBs for a long-duration output of 1MWh. Benefiting from the excellent output power in PP membranes-based Zn-IS FBs systems, the simulated Zn-IS stack (total 0.636013 \$ million) presented a lower price in a 1MWh energy storage system than N117-based Zn-I FBs stack (total 3.184189 \$ million).

Following your suggestion, we have included the cost of chemicals. The discussions have been provided on **Page 14** in the revised manuscript and the corresponding **Supplementary Table 1 & 2** (**Page S51 & S52**).

Supplementary Table 1 | The components and their costs for a 1 MW zinc-iodine flow battery stack

Component	Price per unit (\$ m^2)	Amount of N117 membrane-based Zn-I FBs (m^2)	Total cost of N117 membrane-based Zn-I FBs (\$)	Amount of PP membrane-based Zn-IS FBs (m^2)	Total cost of PP membrane-based Zn-IS FBs (\$)
Bipolar plates	30	5012	150360	3766	112980
Membrane	PP: 10		2506000		37660
	N117: 500				
Frame	15		75180		56490
Carbon felt	20		100240		75320
Gasket-1	1	4 pieces	4	4 pieces	4

Gasket-2	2	4 pieces	8	4 pieces	8
Total cost for 1 MW Zn-I FBs		2.831792 million (\$)		0.282462 million (\$)	

Supplementary Table 2 | The components and their costs for a 1 MW zinc-iodine flow battery stack with 1 MWh.

Chemicals	Molecular weight (g mol⁻¹)	Price per unit (\$ kg⁻¹)	Amount of chemicals (kg)	The cost of chemicals (\$)
ZnCl₂	136.28	11	6934.1	0.076275 million
ZnI₂	319.22	17	16242.4	0.276122 million
Starch	324.28	0.14	8249.9	0.001154 million
Total cost for 1 MW N117 membrane-based Zn-I FBs with 1 MWh (stack + chemicals)			3.184189 million (\$)	
Total cost for 1 MW PP membrane-based Zn-IS FBs with 1 MWh (stack + chemicals)			0.636013 million (\$)	

Summary response: Thanks for your instructive comments. These comments are all valuable and helpful for revising and improving our manuscript, as well as the important guiding significance to our research. We have studied the comments carefully and made corrections, and we hope to meet with approval from you. Revised portions are highlighted in yellow in the revised manuscript. test

Sincerely,

Chunyi Zhi

Department of Materials Science & Engineering

City University of Hong Kong

Email: cy.zhi@cityu.edu.hk

REVIEWER COMMENTS

Reviewer #1 (Remarks to the Author):

The author has addressed the reviewer's comments well. The manuscript has been improved a lot by addressing reviewer's majority of comments and adding additional experiments and explanations. (This manuscript is basically acceptable for publication in Nature Communications)
Two minor questions are in the attached file.

Reviewer #2 (Remarks to the Author):

The authors have addressed my review comments. One minor suggestion is to put the CV data of the one with and without starch in the same figure for a better comparison in the new Supplementary Fig. 25. I recommend acceptance for publication.

Reviewer #3 (Remarks to the Author):

The authors made a significant effort in addressing the comments raised in the first revision.

While several points have been properly clarifying, the comment 1 remains unclear. The authors must address properly this important point before the manuscript becomes suitable for publication in Nat. Commun.

Comment 1 in revision 1: The internal resistance when ion-selective membrane is used is claimed to be drastically increase with respect to the proposed strategy. As a result, the power density achieved using N117 Nafion membrane was only of 28 mW cm⁻² against 70 mW cm⁻² for 1 M starch. The EIS measurements are used to explain the poor power density. However, the difference for Nafion and PP separators is only 0.16 Ohm cm⁻². At 40 mA cm⁻², the voltage drops induced by such a difference would be 4.5 mV. Thus, charge transfer resistance and diffusion must be the origin. However, the low frequencies resistance including all parameters is even lower than 0.16 Ohm cm⁻², which is due to the presence of starch. Therefore, the differences in power density are not supported by the results. This point must be explained properly.

Response: Thanks for your constructive suggestions. We agree with your assertion that using the EIS measurements to substantiate the improved power density might not be rational. Furthermore, we sincerely apologize for the mistake in the EIS results, where the correct unit for EIS should be Ohm cm² rather than Ohm cm⁻² (Nat. Mater., 2020, 19, 195; Nat Commun, 2022, 13, 3184), as shown in Fig. R1. Additionally, it should be clarified that the power density of Zn-I FBs obtained using starch-based electrolyte with a PP membrane at room temperature is 41.58 mW cm⁻², rather than 70 mW cm⁻². Nevertheless, the EIS results can present the impedance measured by applying alternative currents (AC) within a wide frequency range, while the EIS results could not represent the internal resistance of a flow battery under realistic working conditions by applying direct currents (DC). Different DC current densities were applied on the Zn-I FBs at a 50% SOC to measure the internal resistance of the flow batteries. As shown in Fig. R2, the polarization voltage obtained from charge-discharge based on a short working time (1 min) at different current

densities of 10-40 mA cm⁻² were fitted to calculate the internal resistance values. The internal resistance of Zn-IS FBs with starch using a PP membrane is 4.08 Ω cm², which is remarkably lower compared to Zn-IS FBs without starch using an N117 membrane (10.94 Ω cm²). The internal resistance difference at the cell level was mainly responsible for the power difference of these two types of Zn-IS FBs, while the remaining difference could be assigned to the resistance variations caused by activation polarization and concentration polarizations.

We apologize for not discussing this point in our original manuscript. Following your comment, we have cautiously revised the discussions about the EIS section and added the relevant internal resistance under realistic working conditions for these two types of Zn-IS FBs. Please see the highlighted section on Page 9 in the revised manuscripts and the corresponding Supplementary Fig. 27 & Fig. 28 (Pages S29 & S30).

Response from the referee: the response given by the authors is a bit puzzling. If I understood correctly, the authors concluded that “The internal resistance difference at the cell level was mainly responsible for the power difference of these two types of Zn-IS FBs”. Therefore, the higher power density is not an intrinsic advantage of the proposed strategy, but rather a reproducibility issue. However, the text is still claiming it as an intrinsic advantage. If the differences mainly derive from “the internal resistance difference at the cell level”, the discussion about the power density should be omitted. Or the text should clearly indicate that the differences are due to the internal resistance difference at the cell level, so that no benefit from the use of PP membrane-based Zn-IS FBs could be demonstrated. Importantly, the analysis of cost should be based on the same current density and the same power density since the differences were not able to be attributed to the membrane.

“Particularly, PP membrane-based Zn-IS FBs could deliver a high-power density of 41.58 mW cm⁻², demonstrating higher power compared to N117 membrane-based Zn-I FBs with a relatively low power density of 28.41 mW cm⁻² (Fig. 3c). Electrochemical impedance spectroscopy (EIS) result indirectly validated that Zn-IS FBs using PP membranes exhibit lower impedance compared to N117 membranes-based Zn-I FBs (Supplementary Fig. 27). Moreover, as shown in Supplementary Fig. 28, based on the polarization voltage changes fitted at various direct currents, it could substantiate that the internal resistance of Zn-IS FBs system with starch using porous membranes is smaller compared to FBs without starch using N117 membranes. Therefore, the colloidal catholyte-enabled porous PP membranes could endow superior performance of Zn-I FBs compared with N117-based FBs.”

Responses to Reviewer's Comments

Reviewer #1:

The author has addressed the reviewer's comments well. The manuscript has been improved a lot by addressing reviewer's majority of comments and adding additional experiments and explanations. (This manuscript is basically acceptable for publication in Nature Communications). Two minor questions are in the attached file.

General Response: Thank you. We have carefully studied your comments again and addressed them as follows.

Comment 1: *According to the KIx permeability test, N117 membrane exhibits higher permeability ($P_{N117}=2.6014 \times 10^{-7} \text{ cm}^2 \text{ min}^{-1}$) than 1 M starch case ($P_{1M}=7.8637 \times 10^{-8} \text{ cm}^2 \text{ min}^{-1}$). While in the cell test, the N117 membrane exhibits higher CE (99%>) than the 1 M starch case, especially at low current density region (7.5, 12.5, 22.5 mA/cm²). The reviewer is wondering why the N117 membrane shows higher CE than the 1 M starch case.*

Response: Thanks for your insightful observation and comments. We are sorry for our oversight and mistake. As described in the **Supporting Information** of the first revised version, both UV-visible spectroscopy data and digital photographs showed that the permeability of the PP membrane with 1 M starch (**Fig. R1**) is higher than that of the N117 membrane (**Fig. R2**). Moreover, as mentioned in the first revised manuscripts, the N117 membrane exhibited relatively low polyiodide permeability due to the strong Donnan exclusion effect (*Nat Energy*, 2019, 4, 269). Inspired by the point you noted, therefore, we carefully recalculated the data and found that there was a mistake in our previous calculation. The N117 membrane has a permeability of $3.55385 \times 10^{-8} \text{ cm}^2 \text{ min}^{-1}$ rather than the $2.6014 \times 10^{-7} \text{ cm}^2 \text{ min}^{-1}$, which is lower compared with that of the PP membrane as $7.8637 \times 10^{-8} \text{ cm}^2 \text{ min}^{-1}$. We apologize for our mistake again, and we have corrected the I_x^- permeability data for N117 membranes, as shown in **Fig.**

R3. Accordingly, the N117 membranes delivered a higher CE compared to starch-based Zn-IS FBs using PP membranes (**Fig 3c** in the revised manuscript).

Thanks for your delicate observation on keeping our data consistent. We are sorry for our mistake in calculation, and we have corrected it, accordingly, as highlighted in green in our second revised manuscript and **Supplementary Fig. 31 (Page S33)**.

Fig. R1 | The first revised version: the permeability of PP membrane with starch. (a) Photographs of the KI_x permeate solutions through the PP membranes under 2 M KI_x . (b) UV-vis of the KI_x permeated side. (c) $-\ln(1-C_B/C_A)$ vs. permeation time for the determination of permeability of KI_x through N117 membranes under 2 M KI_x . The fits in the $-\ln(1-C_B/C_A)$ vs. t (time) plots in (c) were obtained by linear fitting.

Fig. R2 | The first revised version: the wrong permeability of the N117 membrane. (a) Photographs of the KI_x permeate solutions through N117 membranes under 2 M KI_x . (b) UV-vis of the KI_x permeated side. (c) $-\ln(1-c_B/c_A)$ vs. permeation time for the determination of permeability of KI_x through N117 membranes under 2 M KI_x . The fits in the $-\ln(1-c_B/c_A)$ vs. t (time) plots in (c) were obtained by linear fitting.

Fig. R3 | The current version: the correct permeability of the N117 membrane. (a) Photographs of the KI_x permeate solutions through N117 membranes under 2 M KI_x . (b) UV-vis of the KI_x permeated side. (c) $-\ln(1-c_B/c_A)$ vs. permeation time for the determination of permeability of KI_x through N117 membranes under 2 M KI_x . The fits in the $-\ln(1-c_B/c_A)$ vs. t (time) plots in (c) were obtained by linear fitting.

Comment 2: *The reviewer recommends performing a cell test: 7.5 mA cm^{-2} (10 cycles) \rightarrow 15 mA cm^{-2} (10 cycles) \rightarrow 22.5 mA cm^{-2} (10 cycles) \rightarrow 30 mA cm^{-2} (10 cycles) \rightarrow 37.5 mA cm^{-2} (10 cycles) \rightarrow 7.5 mA cm^{-2} (10 cycles). The reviewer is wondering that the starch case has reversibility for the cell test. Also, the author should suggest the capacity retention data (charge or discharge capacity) according to the cycle test (7.5 to 37.5 mA/cm^2) for both the N117 case and the starch case.*

Response: We are so sorry for not including the data for the current density returning to 7.5 mA cm^{-2} in the original version. Following your suggestion, we have supplemented the rate performance data under both room and high temperatures to evaluate the reversibility of starch-based FBs further. In particular, as shown in **Fig. R4 & R5**, the flow-mode Zn-IS FBs using the PP membrane exhibited superior rate cycling and performance with high CE, VE and EE of 94-91-86, 96-85-83%, 98-81-78%, 98.5-75-74% and 98.6-70-70% at 7.5, 15, 22.5, 30 and 37.5 mA cm^{-2} , respectively. Moreover, after switching the

current density back to 7.5 mA cm⁻², the EE of Zn-IS FBs was restored to 86%, indicating the excellent reversible ability of colloidal starch-based cells at various rates. Similarly, the as-developed Zn-IS FBs also revealed superior rate reversibility under high-temperature (50 °C) conditions (**Fig. R6 & R7**).

On the other hand, as per your suggestion, we have supplemented the capacity retention data (charge or discharge capacity) in the rate cycle test (7.5 to 3.75 mA cm⁻²). Notably, the discharge capacity was directly shown for comparison, where the charging process was terminated by the same charging capacity of 30 mAh. As shown in **Fig. R8**, it showed similar discharge capacity retention at different current densities (7.5 to 3.75 mA cm⁻²) for the N117 membranes-based Zn-I FBs (29.7, 29.8, 29.8, 29.8 mAh) and the as-developed Zn-IS FBs (28.2, 28.8, 29.4, 29.6, 29.6 mAh) in this work.

We have supplied the rate performance back to 7.5 mA cm⁻² under both room (25 °C) and high-temperature (50 °C) conditions on **Pages 9-12** in the second revised manuscript and the corresponding **Supplementary Fig. 23, Fig. 29 & Fig. 44 (Pages S25, S31 & S46)**, respectively.

Fig. R4 | Electrochemical performance of the Zn-IS FBs at 25 °C. (a) CE, VE and EE of the Zn-IS FBs cell (2 ml of 2 M ZnI₂ with 1 M starch || PP membrane || 8 ml of 2 M ZnCl₂, 4 cm² membrane area) under 7.5, 15, 22.5, 30, 37.5 and 7.5 mA cm⁻², (b) Voltage profiles of the Zn-IS FBs under 7.5, 15, 22.5,

30 and 37.5 mA cm⁻². (c) The polarization of the Zn-IS FBs using different membranes (PP with 1 M starch & N117 without starch). (d) Cycling performances of Zn-I FBs flow-cell system at high current density (30 mA cm⁻²). (e) Cycling performances of Zn-IS FBs flow-cell system with/without starch at high volume capacity (50% SOC, 36 Ah L⁻¹) under a current density of 22.5 mA cm⁻². The inset in (d) & (e) are the corresponding voltage profiles.

Fig. R5 | Galvanostatic cycling of Zn-IS FBs (2 ml of 2 M ZnI₂ with 1 M starch || PP membrane || 8 ml of 2 M ZnCl₂, 4 cm² membrane area) under 7.5, 15, 22.5, 25, 30 and 7.5 mA cm⁻² at high temperature (25 °C).

Fig. R6 | Electrochemical performance of the Zn-IS FBs at 50 °C. (a) CE, VE and EE of the Zn-IS FBs cell (2 ml of 2 M ZnI₂ with 1 M starch || PP membrane || 8 ml of 2 M ZnCl₂, 4 cm² membrane area) under 7.5, 15, 22.5, 30, 37.5 and 7.5 mA cm⁻², (b) Voltage profiles of the Zn-IS FBs under 7.5, 15, 22.5, 30 and 37.5 mA cm⁻². (c) The polarization of the Zn-IS FBs using different membranes (PP with 1 M starch & N117 without starch). (d) Cycling performances of Zn-I FBs flow-cell system at high current density (30 mA cm⁻²). (e) Cycling performances of Zn-IS FBs flow-cell system with/without starch at high volume capacity (50% SOC, 36 Ah L⁻¹) under a current density of 22.5 mA cm⁻². The inset in (d) & (e) are the corresponding voltage profiles.

Fig. R7 | Galvanostatic cycling of Zn-IS FBs (2 ml of 2 M ZnI₂ with 1 M starch || PP membrane || 8 ml of 2 M ZnCl₂, 4 cm² membrane area) under 7.5, 15, 22.5, 25, 30 and 7.5 mA cm⁻² at high temperature (50 °C).

Fig. R8 | Discharging capacity of the Zn-I FBs flow cell using different membranes (PP with 1 M starch & N117 without starch) during charging to 30 mAh.

Responses to Reviewer's Comments

Reviewer #2:

The authors have addressed my review comments. One minor suggestion is to put the CV data of the one with and without starch in the same figure for a better comparison in the new Supplementary Fig. 25. I recommend acceptance for publication.

Response: Thanks for your professional assessment of our work. We appreciate your insightful assessments regarding the improvements made for cost-effective Zn-IS flow batterie. We have carefully reviewed your comments and revised our manuscript to address them.

As per your comment, the CV curve of iodides-based cathode with and without starch was put in the same figure for a better comparison, as shown in **Fig. R1**. Meanwhile, we have supplied this data in the second revised manuscript and the corresponding **Supplementary Fig. 25 (Page S27)**.

Fig. R1 | (a) The photograph of the reactor under flowing-mode conditions. Cyclic voltammetry of redox reactions in (b) 0.1 M KI without and with 1 M starch under flowing-mode conditions. (c) the corresponding Tafel plots in different electrolytes.

Responses to Reviewer's Comments

Reviewer #3:

The authors made a significant effort in addressing the comments raised in the first revision.

While several points have been properly clarifying, the comment 1 remains unclear. The authors must address properly this important point before the manuscript becomes suitable for publication in Nat. Commun.

Comment 1 in the First revision: *The internal resistance when ion-selective membrane is used is claimed to be drastically increase with respect to the proposed strategy. As a result, the power density achieved using N117 Nafion membrane was only of 28 mW cm⁻² against 70 mW cm⁻² for 1 M starch. The EIS measurements are used to explain the poor power density. However, the difference for Nafion and PP separators is only 0.16 Ohm cm⁻². At 40 mA cm⁻², the voltage drops induced by such a difference would be 4.5 mV. Thus, charge transfer resistance and diffusion must be the origin. However, the low frequencies resistance including all parameters is even lower than 0.16 Ohm cm⁻², which is due to the presence of starch. Therefore, the differences in power density are not supported by the results. This point must be explained properly.*

Response in the First revision: *Thanks for your constructive suggestions. We agree with your assertion that using the EIS measurements to substantiate the improved power density might not be rational. Furthermore, we sincerely apologize for the mistake in the EIS results, where the correct unit for EIS should be Ohm cm² rather than Ohm cm⁻² (Nat. Mater., 2020, 19, 195; Nat Commun, 2022, 13, 3184), as shown in Fig. R1. Additionally, it should be clarified that the power density of Zn-I FBs obtained using starch-based electrolyte with a PP membrane at room temperature is 41.58 mW cm⁻², rather than 70 mW cm⁻². Nevertheless, the EIS results can present the impedance measured by applying alternative currents (AC) within a wide frequency range, while the EIS results could not represent the internal resistance of a flow battery under realistic working conditions by applying direct currents (DC).*

Different DC current densities were applied on the Zn-I FBs at a 50% SOC to measure the internal resistance of the flow batteries. As shown in Fig. R2, the polarization voltage obtained from charge-discharge based on a short working time (1 min) at different current densities of 10-40 mA cm⁻² were fitted to calculate the internal resistance values. The internal resistance of Zn-IS FBs with starch using a PP membrane is 4.08 Ω cm², which is remarkably lower compared to Zn-IS FBs without starch using an N117 membrane (10.94 Ω cm²). The internal resistance difference at the cell level was mainly responsible for the power difference of these two types of Zn-IS FBs, while the remaining difference could be assigned to the resistance variations caused by activation polarization and concentration polarizations.

We apologize for not discussing this point in our original manuscript. Following your comment, we have cautiously revised the discussions about the EIS section and added the relevant internal resistance under realistic working conditions for these two types of Zn-IS FBs. Please see the highlighted section on Page 9 in the revised manuscripts and the corresponding Supplementary Fig. 27 & Fig. 28 (Pages S29 & S30).

Response from the referee: *the response given by the authors is a bit puzzling. If I understood correctly, the authors concluded that “The internal resistance difference at the cell level was mainly responsible for the power difference of these two types of Zn-IS FBs”. Therefore, the higher power density is not an*

intrinsic advantage of the proposed strategy, but rather a reproducibility issue. However, the text is still claiming it as an intrinsic advantage. If the differences mainly derive from “the internal resistance difference at the cell level”, the discussion about the power density should be omitted. Or the text should clearly indicate that the differences are due to the internal resistance difference at the cell level, so that no benefit from the use of PP membrane-based Zn-IS FBs could be demonstrated. Importantly, the analysis of cost should be based on the same current density and the same power density since the differences were not able to be attributed to the membrane.

“Particularly, PP membrane-based Zn-IS FBs could deliver a high-power density of 41.58 mW cm^{-2} , demonstrating higher power compared to N117 membrane-based Zn-I FBs with a relatively low power density of 28.41 mW cm^{-2} (Fig. 3c). Electrochemical impedance spectroscopy (EIS) result indirectly validated that Zn-IS FBs using PP membranes exhibit lower impedance compared to N117 membranes-based Zn-I FBs (Supplementary Fig. 27). Moreover, as shown in Supplementary Fig. 28, based on the polarization voltage changes fitted at various direct currents, it could substantiate that the internal resistance of Zn-IS FBs system with starch using porous membranes is smaller compared to FBs without starch using N117 membranes. Therefore, the colloidal catholyte-enabled porous PP membranes could endow superior performance of Zn-I FBs compared with N117-based FBs.”

Comment: *the response given by the authors is a bit puzzling. If I understood correctly, the authors concluded that “The internal resistance difference at the cell level was mainly responsible for the power difference of these two types of Zn-IS FBs”. Therefore, the higher power density is not an intrinsic advantage of the proposed strategy, but rather a reproducibility issue. However, the text is still claiming it as an intrinsic advantage. If the differences mainly derive from “the internal resistance difference at the cell level”, the discussion about the power density should be omitted. Or the text should clearly indicate that the differences are due to the internal resistance difference at the cell level, so that no benefit from the use of PP membrane-based Zn-IS FBs could be demonstrated. Importantly, the analysis of cost should be based on the same current density and the same power density since the differences were not able to be attributed to the membrane.*

Response: Thanks for your professional evaluation of our work. We are very sorry for the confusing expressions. You are correct that the low internal resistance in the FBs leads to high power density.

On the other hand, it should be noted that, in our work, the low internal resistance of the FBs is the result of our strategy, that is, the combination of porous PP membranes and colloidal starch-based iodine catholytes. Specifically, the PP membrane plays a crucial role in low internal resistance due to the properties of porous and larger pore sizes, but it deteriorates the stability of the FBs due to the cross-over. Therefore, we used colloidal starch electrolytes as the iodine catholyte. Then, the PP membrane-based Zn-IS FBs can retain the low internal resistance and effectively suppress cross-over.

In summary, our strategy, that is, the combination of PP membrane and starch electrolyte, results in low resistance and good stability, and low resistance leads to high power density. In contrast, blank Zn-I FBs using the dense N117 membrane with smaller pore sizes exhibited higher internal resistance, leading to a relatively low power density.

Moreover, we are very sorry for not clearly explaining the two internal resistances provided in the first revised manuscript, and we have re-analyzed the data.

The first one is the impedance calculated using the EIS results. The device applied **alternating currents (AC)** within a certain **AC** frequency range.

The second one is the resistance tested by the electrochemical methods. The device was applied with **direct currents (DC)**.

Under the **AC** condition, this measurement assesses the impedance of the battery, which mainly reflects more intricate information relating to charge interface transport, interface chemical reactions and so on. For the **DC** mode, this method evaluates the internal resistance of the battery under actual working conditions, which is more meaningful to determine the power density of the battery.

The internal resistance under **DC** conditions was further analyzed here. To calculate the internal resistance of the FB during the charging/discharging process, we measured the overpotentials ($U_2 - U_1$) when applying a specific magnification current (I). The resistance could be calculated following $R = (U_2 - U_1) / 2I$ (*Environ. Sci. Technol.* 2016, **50**, 9791; *J. Electrochem. Soc.*, 1965, **112**, 657). It should be noted that the charge and discharge operation here lasted for a short period of time (1 min) to keep the 50% state of the charge state of the polyiodide catholyte. Specifically, as shown in **Fig. R1**, the calculated internal resistance of Zn-IS FBs with starch using a PP membrane was $4.08 \Omega \text{ cm}^2$, which was lower than Zn-I FBs without starch using the N117 membrane ($10.94 \Omega \text{ cm}^2$).

Thus, the difference ($6.86 \Omega \text{ cm}^2$) in internal resistance measured at 50% SOC of these two batteries could cause the difference of 9.65 mW cm^2 in power density at 37.5 mA cm^{-2} , almost aligned to the measured difference (13.17 mW cm^2 at 37.5 mA cm^{-2} , calculated by **Fig. 3c** in the manuscript) in the power density of the discharge process. Notably, a small gap in the power density differences should be caused by the concentration and activation polarization in the FBs. In other words, the difference in power density between PP membrane-based Zn-IS FBs with starch and N117 membrane-based Zn-I FBs without starch in **Fig. 3c** (in the manuscript) is reasonable.

Based on the above data, we analyzed again the cost of two types of FBs under the same current density. We consider a 1 MW Zn-I FBs stack with 1 MWh as an example. The developed PP membrane-based Zn-IS FBs (3766 m^2) need less reaction area than the N117 membranes-based Zn-I FBs (5012 m^2),

which is due to the higher power density of PP-based systems (26.55 mW cm^{-2}) than N117-based systems (19.95 mW cm^{-2}). Therefore, the Zn-IS FBs stack uses PP membranes with less area and lower cost (total \$636,013.00) than the Zn-I FBs stack using N117 membranes with more area and higher cost (total \$3,184,189.00), leading to a lower installation cost. The detailed cost and the amount of components for the 1 MW flow cell stack with 1 MWh were provided in **Supplementary Table 1 & Table 2**.

Again, we are sorry for our confusing expressions. We have corrected them and added more explanations in the revised manuscript (**Page 9**) and the corresponding supporting information (**Supplementary Fig. 28 (Page S30)**).

Fig. R1 | The polarization voltage of the Zn-I FBs flow cell using different membranes ((a) PP membranes with 1 M starch & (b) N117 membranes without starch) under charging to 50% SOC at different current density (10 - 40 mA cm^{-2}).

Summary response: Thanks for your instructive comments. These comments are all valuable and helpful for revising and improving our manuscript, as well as the important guiding significance to our research. We have studied the comments carefully and made corrections. Revised portions are highlighted in **yellow (the first revised version) and green (the second revised version)** in the revised manuscript.

Sincerely,

Chunyi Zhi

Department of Materials Science & Engineering

City University of Hong Kong

Email: cy.zhi@cityu.edu.hk

REVIEWERS' COMMENTS

Reviewer #1 (Remarks to the Author):

Now, I think it can be published as it is.

Reviewer #2 (Remarks to the Author):

The authors have addressed my comments adequately. I recommend acceptance.

Reviewer #3 (Remarks to the Author):

The authors have addressed the remaining point. Thus, I recommend now the publication of this work in Nat. Commn.

Department of Materials Science and Engineering
Professor, Dr Chunyi ZHI

City University of Hong Kong
83 Tat Chee Avenue, Kowloon
Hong Kong SAR, China
Tel: +852 3442 7891
Fax: +852 3442 0538
Email: cy.zhi@cityu.edu.hk

Responses to Reviewers

Dear Reviewer,

Thanks a lot for your constructive comments about our manuscript entitled “*Starch-mediated Colloidal Chemistry for Highly Reversible Zinc-based Polyiodide Redox Flow Batteries*” (NCOMMS-23-38205B), which we really appreciate. The comments were addressed point-by-point below.

Reviewer #1:

Now, I think it can be published as it is.

Response: We sincerely appreciate the reviewer’s comments, which can improve our manuscript.

Reviewer #2:

The authors have addressed my comments adequately. I recommend acceptance.

Response: We are sincerely grateful for the reviewer’s comments, which can advance our manuscript.

Reviewer #3:

The authors have addressed the remaining point. Thus, I recommend now the publication of this work in Nat. Commn..

Response: We sincerely thank the reviewer’s comments, and we believe it can improve our manuscript significantly.

Sincerely,

Chunyi Zhi

Department of Materials Science & Engineering

City University of Hong Kong

Email: cy.zhi@cityu.edu.hk